# Trust-Region Twisted Policy Improvement

**Joery A. de Vries** [1]  **Jinke He** [1]  **Yaniv Oren** [1]  **Matthijs T. J. Spaan** [1]

## Abstract

Monte-Carlo tree search (MCTS) has driven many recent breakthroughs in deep reinforcement learning (RL). However, scaling MCTS to parallel compute has proven challenging in practice which has motivated alternative planners like sequential Monte-Carlo (SMC). Many of these SMC methods adopt particle filters for smoothing through a reformulation of RL as a policy inference problem. Yet, persisting design choices of these particle filters often conflict with the aim of online planning in RL, which is to obtain a policy improvement at the start of planning. Drawing inspiration from MCTS, we tailor SMC planners specifically to RL by improving data generation within the planner through constrained action sampling and explicit terminal state handling, as well as improving policy and value target estimation. This leads to our *Trust-Region Twisted* SMC (TRT-SMC), which shows improved runtime and sample-efficiency over baseline MCTS and SMC methods in both discrete and continuous domains.

## 1. Introduction

Monte-Carlo tree search (MCTS) with neural networks (Browne et al., 2012) has enabled many recent successes in sequential decision making problems such as board games (Silver et al., 2018), Atari (Ye et al., 2021), and algorithm discovery (Fawzi et al., 2022; Mankowitz et al., 2023). These successes demonstrate that combining decision-time planning and reinforcement learning (RL) often outperforms methods that utilize search or deep neural networks in isolation. Naturally, this has stimulated many studies to understand the role of planning in combination with learning (Guez et al., 2019; de Vries et al., 2021; Hamrick et al., 2021; Bertsekas, 2022; He et al., 2024) along with studies to improve specific aspects of the base algorithm (Hubert et al., 2021; Danihelka et al., 2022; Antonoglou et al., 2022; Oren et al., 2025).

Although MCTS has been a leading technology in recent breakthroughs, the tree search is inherently sequential, can deteriorate agent performance at small planning budgets (Grill et al., 2020), and requires significant modifications for general use in RL (Hubert et al., 2021). The sequential nature of MCTS is particularly crippling as it limits the full utilization of modern hardware such as GPUs. Despite follow-up work attempting to address specific issues, alternative planners have since been explored that avoid these flaws. Successful alternatives in this area are often inspired by stochastic control (Del Moral, 2004; Åström, 2006), examples include path integral control (Theodorou et al., 2010; Williams et al., 2015; Hansen et al., 2022) and related sequential Monte-Carlo (SMC) methods (Naesseth et al., 2019; Chopin & Papaspiliopoulos, 2020).

Specifically, recent variational SMC planners (Naesseth et al., 2018; Macfarlane et al., 2024) have shown great potential in terms of generality, performance, and scalability to parallel compute. These methods adopt a particle filter for trajectory smoothing to enable planning in RL (Piché et al., 2019). However, the distribution of interest for these particle filters do not perfectly align with learning and exploration for RL agents. Namely, recent SMC methods focus on estimation of the trajectory distribution under an unknown policy, and not the actual unknown policy at the state where we initiate the planner. We find that this mismatch can cause SMC planners to suffer from unnecessarily high-variance estimation and waste much of their compute and data during planning. In other words, online planning in RL should serve as a local *approximate policy improvement* (Chan et al., 2022; Sutton & Barto, 2018). Fortunately, existing MCTS and SMC literature provides various directions to achieve this (Moral et al., 2010; Svensson et al., 2015; Lawson et al., 2018; Danihelka et al., 2022; Grill et al., 2020), but their use in SMC-planning remains largely unrealized.

This paper aims to address the current limitations of bootstrapped particle filter planners for RL by drawing inspiration from MCTS. Our contributions are 1) to make more accurate estimates of statistics extracted by the planner, at the start of planning, and 2) enhancing data-efficiency inside the planner. We address the pervasive *path-degeneracy* prob-

---

[1]Delft University of Technology, Delft, the Netherlands. Correspondence to: Joery A. de Vries <J.A.deVries@tudelft.nl>.

*Proceedings of the 42$^{nd}$ International Conference on Machine Learning*, Vancouver, Canada. PMLR 267, 2025. Copyright 2025 by the author(s).

lem in SMC by backing up accumulated reward and value data to perform policy inference and construct value targets. Next, to reduce the variance of estimated statistics due to re-sampling, we use exponential *twisting* functions to improve the sampling trajectories inside the planner. We also impose adaptive trust-region constraints to a prior policy, to control the bias-variance trade-off in sampling proposal trajectories. Finally, we modify the resampling method (Naesseth et al., 2019) to correct particles that become permanently stuck in absorbing states due to termination in the baseline SMC. We dub our new method *Trust-Region Twisted* SMC and demonstrate improved sample-efficiency and runtime scaling over SMC and MCTS baselines in discrete and continuous domains.

## 2. Background

We want to find an optimal policy $\pi$ for a sequential decision-making problem, which we formalize as an infinite-horizon Markov decision process (MDP) (Sutton & Barto, 2018). We define states $S \in \mathcal{S}$, actions $A \in \mathcal{A}$, and rewards $R \in \mathbb{R}$ as random variables, where we write $H_{1:T} = \{S_t, A_t\}_{t=1}^T$ as the joint random variable of a *finite* sequence,

$$p_\pi(H_{1:T}) = \prod_{t=1}^T \pi(A_t|S_t)p(S_t|S_{t-1}, A_{t-1}), \quad (1)$$

where $p(S_1|A_0, S_0) \triangleq p(S_1)$ is the initial state distribution, $p(S_{t+1}|S_t, A_t)$ is the transition model, and $\pi(A_t|S_t)$ is the policy. We denote the set of admissible policies as $\Pi \triangleq \{\pi|\pi : \mathcal{S} \to \mathbb{P}(\mathcal{A})\}$, our aim is to find a parametric $\pi_\theta \in \Pi$ (e.g., a neural network) such that $\mathbb{E}_{p_{\pi_\theta}(H_{1:T})}[\sum_{t=1}^T R_t]$ is maximized, where we abbreviate $R_t = R(S_t, A_t)$ for the rewards. For convenience, we subsume the discount factor $\gamma \in [0, 1]$ into the transition probabilities as a termination probability of $p_{\text{term}} = 1 - \gamma$ assuming that the MDP always ends up in an absorbing state $S_T$ with zero rewards.

### 2.1. Control as Inference

The reinforcement learning problem can be recast as a probabilistic inference problem through the control as inference framework (Levine, 2018). This reformulation has lead to successful algorithms like MPO (Abdolmaleki et al., 2018) that naturally allow *regularized* policy iteration (Geist et al., 2019), which is highly effective in practice with neural network approximation. Additionally, it enables us to directly use tools from Bayesian inference on our graphical model.

To formalize this, the distribution for $H_{1:T}$ can be conditioned on a binary outcome variable $\mathcal{O}_t \in \{0, 1\}$, then given a likelihood for $p(\mathcal{O}_{1:T} = 1|H_{1:T}) = p(\mathcal{O}_{1:T}|H_{1:T})$ this gives rise to a posterior distribution $p(H_{1:T}|\mathcal{O}_{1:T})$. Typically, we use the exponentiated sum of rewards for the (unnormalized) likelihood $p(\mathcal{O}_{1:T}|H_{1:T}) \propto \prod_{t=1}^T \exp R_t$.

**Definition 2.1.** The posterior factorizes as

$$p_\pi(H_{1:T}|\mathcal{O}_{1:T}) = \prod_{t=1}^T p_\pi(A_t|S_t, \mathcal{O}_{t:T})p(S_t|S_{t-1}, A_{t-1}),$$

assuming that $S_{t+1} \perp\!\!\!\perp \mathcal{O}_{1:T}|S_t, A_t$ and $A_t \perp\!\!\!\perp \mathcal{O}_{<t}|S_t$.

The key part of Definition 2.1 is the posterior policy $p_\pi(A_t|S_t, \mathcal{O}_{t:T})$ which, using Bayes rule, reads as

$$p_\pi(A_t|S_t, \mathcal{O}_{t:T}) \propto \pi(A_t|S_t) \exp[\ln p(\mathcal{O}_{t:T}|S_t, A_t)]. \quad (2)$$

This connects us to an (expected-reward) maximum-entropy reinforcement learning setting (Toussaint, 2009; Ziebart, 2010), since the exponent admits a soft-Bellman recursion,

$$\ln p(\mathcal{O}_{t:T}|S_t, A_t) = R(S_t, A_t) + \ln \mathbb{E}\, e^{V_{soft}^\pi(S_{t+1})}, \quad (3)$$

where the expectation is over the dynamics $p(S_{t+1}|S_t, A_t)$. An important property of the posterior policy is that it is not an optimal policy in the traditional objective $\mathbb{E}_{p_\pi}[\sum_t R_t]$, but only provides an improvement over a prior policy $\pi$.

**Theorem 2.2.** *The posterior policy* $p(A_t|S_t, \mathcal{O}_{t:T}), \forall t$, *is the optimal policy* $q^*$ *for the regularized MDP,*

$$q^* = \arg\max_{q \in \Pi} \mathbb{E}\left[\sum_{t=1}^T R_t - KL(q(a|S_t)\|\pi(a|S_t))\right],$$

*where* $q^*$ *guarantees a policy improvement in the unregularized MDP,* $\mathbb{E}_{p_{q^*}(H_{1:T})}[\sum_{t=1}^T R_t] \geq \mathbb{E}_{p_\pi(H_{1:T})}[\sum_{t=1}^T R_t]$.

*Proof.* See Appendix A.3 or Sec. 2.4 of (Levine, 2018). □

The optimal regularized policy $q^*$ can be interpreted as a *variational* policy[1] that minimizes the evidence gap (Bishop, 2007), or conversely, maximizes the evidence lower-bound to $\ln p(\mathcal{O}_{1:T})$. In practice, this can be used in expectation-maximization (EM) methods (Neal & Hinton, 1998) to iteratively update the prior policy $\pi_{(n+1)} \leftarrow q^*_{(n)}$. This gives rise to an effective framework for approximate policy iteration that is both amenable to gradient based updating of $\pi_\theta$ and provably recovers the traditional (locally) optimal policy $\max_{\pi \in \Pi} \mathbb{E}_{p_\pi}[\sum_t R_t]$ as $n \to \infty$ (see Appendix A).

### 2.2. Particle Filter Planning for RL

Although the estimation of the regularized optimal policy $q^*$ mitigates a number of practical challenges in deep RL, it also requires re-estimating $q^*_{(n)}$ after each update to the prior $\pi_{(n+1)}$. Additionally, the posterior policy for any $\pi$ always requires the solutions to the soft value-estimation problem for $Q^\pi_{soft}$ and $V^\pi_{soft}$. Algorithms like MPO deal

---

[1] From this point on we will interchange the regularized optimal policy $q^*_{(n)}(A_t|S_t)$ with the posterior policy $p_{\pi_{(n)}}(A_t|S_t, \mathcal{O}_{t:T})$.

with this by approximating the value using neural networks $Q_\theta^\pi \approx Q_{soft}^\pi$ to then estimate the posterior policy through Monte-Carlo. Intuitively, this can be interpreted as a 1-timestep approximation to $q^*$. To see this, evaluating $Q_\theta^\pi(S_t, A_t)$ over samples $A_t \sim \pi(A_t|S_t)$ amortizes the message-passing process for evaluating $Q_{soft}^\pi$ into a direct (cheap) mapping from $\mathcal{S} \times \mathcal{A} \to \mathbb{R}$. The main limitation of this approach is the bias induced by this new function $Q_\theta^\pi$. Towards this end, sequential Monte-Carlo (SMC) methods (Piché et al., 2019; Naesseth et al., 2019) offer a powerful model-based strategy for improving the estimate of $q^*$ through a multi-timestep.

The SMC algorithm for RL (Piché et al., 2019) is a sequential importance sampling (IS) method that aims to draw samples $H_{1:t}$ from the posterior through a proposal distribution $p_q(H_{1:t})$ (as in Eq. 1 for some $q \in \Pi$). By definition, our graphical model (Def. 2.1) factorizes recursively,

$$p_\pi(H_{1:T}|\mathcal{O}_{1:T}) = p_\pi(H_{1:t}|\mathcal{O}_{1:T})p_\pi(H_{t+1:T}|H_t, \mathcal{O}_{t+1:T}), \quad (4)$$

meaning that we can sample data from $H_{1:t} \sim p_q(H_{1:t})$ and accumulate the IS-weights sequentially.

**Corollary 2.3** (Sequential importance sampling). *Assuming access to the transition model $p(S_{t+1}|S_t, A_t)$, we obtain the importance sampling weights for $p_\pi(H_{1:t}|\mathcal{O}_{1:T})/p_q(H_{1:t})$,*

$$w_t = w_{t-1} \cdot \frac{\pi(A_t|S_t)}{q(A_t|S_t)} \exp(R_t) \frac{\mathbb{E}[\exp V_{soft}^\pi(S_{t+1})]}{\exp V_{soft}^\pi(S_t)}.$$

*Proof.* The dynamics terms in the weights cancel out, the rest follows by definition from Equations 2, 3, and 4. □

In practice we cannot realistically compute $w_t$ since it requires the soft-values $V_{soft}^\pi$. However, we can again approximate this $\widetilde{w}_t \approx w_t$ with the amortized estimate $V_\theta^\pi \approx V_{soft}^\pi$ at every intermediate timestep (Pitt & Shephard, 1999; Lawson et al., 2018). With this, we can estimate posterior statistics using a single forward pass through: sampling traces $H_{1:t}^{(i)}$, accumulating their approximate weights $\widetilde{w}_t^{(i)}$, and then normalizing these $\overline{w}_t^{(i)} = \frac{\widetilde{w}_t^{(i)}}{\sum_j \widetilde{w}_t^{(j)}}$ to obtain

$$\mathbb{E}_{p_\pi(H_{1:t}|\mathcal{O})} f(H_{1:t}) = \mathbb{E}_{p_q}[w_t \cdot f(H_{1:t})] \quad (5)$$
$$\approx \sum_{i=1}^K \overline{w}_t^{(i)} f(H_{1:t}^{(i)}), \quad H_{1:t}^{(i)} \sim p_q,$$

where $f : \mathcal{H}^t \to \mathcal{Y}$ is some arbitrary function over the data. For instance, the statistic $f(H_{1:t}) = \sum_{j=1}^t R_j$ would yield an estimate of the expected finite-horizon sum of rewards.

---

**Algorithm 1** Bootstrapped Particle Filter for RL

**Require:** $K$ (number of particles), $m$ (depth)
1: Initialize:
   - Ancestor identifier $\{J_1^{(i)} = i\}_{i=1}^K$
   - States $\{S_1^{(i)} \sim p(S_1)\}_{i=1}^K$
   - Weights $\{\widetilde{w}_0^{(i)} = 1\}_{i=1}^K$
2: **for** $t = 1$ to $m$ **do**
   // Update particles
3: $\quad \{A_t^{(i)} \sim q(A_t|S_t^{(i)})\}_{i=1}^K$
4: $\quad \{S_{t+1}^{(i)} \sim p(S_{t+1}|S_t^{(i)}, A_t^{(i)})\}_{i=1}^K$
5: $\quad \{\widetilde{w}_t^{(i)} = \widetilde{w}_{t-1}^{(i)} \frac{\pi(A_t^{(i)}|S_t^{(i)})}{q(A_t^{(i)}|S_t^{(i)})} e^{R_t^{(i)}} \frac{\mathbb{E} \exp V_\theta^\pi(S_{t+1}^{(i)})}{\exp V_\theta^\pi(S_t^{(i)})}\}_{i=1}^K$
   // Bootstrap (periodically) through resampling
6: $\quad \{(J_t^{(i)}, S_{t+1}^{(i)}, A_t^{(i)})\}_{i=1}^K \sim \text{Multinomial}(K, \overline{w}_t)$
7: $\quad \{\widetilde{w}_t^{(i)} = 1\}_{i=1}^K$
8: **end for**
9: **return** $\{J_{1:m}^{(i)}, H_{1:m}^{(i)}, \widetilde{w}_{1:m}^{(i)}\}_{i=1}^K$

---

**Bootstrapped Filter** The bootstrapped particle filter for RL, as proposed by Piché et al. (2019), improves the estimation in Eq. 5 through (periodic) *resampling*. This mitigates the issue of weight-impoverishment in sequential-IS, where some normalized weights dominate others $\overline{w}_t^{(i)} \gg \overline{w}_t^{(j)}, j \neq i$. A common strategy for this is multinomial resampling (Chopin & Papaspiliopoulos, 2020), as shown in Algorithm 1. This method samples a number of traces $H_{1:m}^{(i)} \sim p_q, i \in [1, K]$, referred to as particles, and periodically drops or duplicates these samples based on their weights $\overline{w}_{1:m}^{(i)}, m \in [1, t]$, before resetting their weights back to a uniform distribution (bootstrap). Prior work (Piché et al., 2019; Macfarlane et al., 2024) then estimates the policy $\hat{q}^*$ as a weighted mixture of point-masses,

$$\hat{q}^*(A_t = a|S_t) = \sum_{i=1}^K \overline{w}_{t+m}^{(i)} \delta(A_t^{(J_{t+m}^{(i)})} = a), \quad (6)$$

where $J_{t+m}^{(i)} \in [1, K]$ is the index tracking each samples' ancestor at the start of planning $t$ and $\delta(\cdot)$ is a Dirac delta function over $\mathcal{A}$ for the ancestor action-particles.

A key property of Algorithm 1 is that it can estimate $\hat{q}^*$ through a single forward pass of $K$ particles from $t$ to $t + m$ (Moral et al., 2010). This allows for parallel sampling and updating of particles, with communication only required during the resampling step. As a result, it scales efficiently on modern GPU hardware, with memory complexity of $\mathcal{O}(K)$ and a parallelized time complexity of $\mathcal{O}(m)$.

**Variational SMC** Intuitively, resampling improves our estimate of Eq. 5 by 'correcting' particles with low-likelihood

to higher likelihood regions of the target distribution. However, resampling also introduces noise and would ideally not be needed (or to a lesser extent) if our proposal distribution $p_q$ produced samples that better match our target. To this end, variational SMC methods (Naesseth et al., 2018; Gu et al., 2015) learn a proposal distribution $q_\theta$ from sample estimates of the posterior target $\mathbb{E}[\hat{q}^*] = q^*$. Macfarlane et al. (2024) use this strategy with neural networks to learn $q_\theta \approx q^*$. Given the neural network iterates $\{\theta_{(n)}\}_{n=1}^N$, their method also updates the prior policy $\pi_{(n+1)} \leftarrow q_{\theta_{(n)}}$ at every step $n$. In other words, they combine variational SMC in an EM-loop (Neal & Hinton, 1998) by using the learned proposals $q_{\theta_{(n)}}$ as the prior to regularize the posterior $p_{\pi_{(n+1)}}(A_t|S_t, \mathcal{O}_{t:T})$.

## 3. Tailoring Particle Filter Planning to RL

A key benefit of online planning in reinforcement learning (RL) is generating locally improved policies at each timestep $t$ compared to a prior policy, which enhances learning targets and diversity of data (Hamrick et al., 2021; Silver et al., 2018). Despite recent progress in improving sequential Monte-Carlo (SMC) methods for local approximate *policy improvement* (Chan et al., 2022), persisting design choices from conventional particle filtering do not align directly with this goal. For instance, we find that the problem of *path degeneracy*, inherent to particle filtering (Svensson et al., 2015), can cause policy inference in Eq. 6 to degenerate with deep search and waste much of the planner's generated data. We also argue that variational SMC planners still make inefficient use of their particle budget due to periodic resampling, which can cause particles to get stuck in terminal states and delays using value information. By contrasting this with Monte-Carlo tree search (MCTS), which specifically focuses on improving the policy at the root (Grill et al., 2020), we tailor the SMC-planner for local *policy improvement*. This preserves the forward-only implementation of particle filtering while incorporating benefits inspired by MCTS.

### 3.1. Adapting the Proposals for Sample-Efficiency

The resampling step in SMC (line 6, Alg. 1) is essential for redistributing particles that are unlikely under the posterior policy. Simultaneously, variational SMC methods (Macfarlane et al., 2024) can shift some of this responsibility to a learned proposal distribution $q_\theta \approx \hat{q}^*$ by generating better samples. This is useful as it reduces variance induced by resampling and amortizes posterior inference (Naesseth et al., 2018). However, in EM-loop style algorithms, the learned proposal distribution $q_\theta$ is an estimate of the posterior policy that is regularized to a prior from a previous iteration $q_\theta \approx p_{\pi_{(n-1)}}(A_t|S_t, \mathcal{O}_{t:T})$, and not to the current iteration $\pi_{(n)}$. This is computationally wasteful with infrequent resampling, as it can take multiple transitions before moving particles to rewarding trajectories (Lioutas et al., 2023).

For this reason, we extend the proposal distribution to more accurately reflect the *next* posterior policy $p_{\pi_{(n)}}(A_t|S_t, \mathcal{O}_{t:T})$ by using both a learned policy $q_\theta \equiv \pi_{(n)}$ and value $\exp Q_\theta^\pi$. This is also known as exponential twisting (Asmussen & Glynn, 2007) of the proposal, which is similarly used in the target distribution (Corollary 2.3). Twisting reduces the variance for $\hat{q}^*$ at the cost of some bias due to $Q_\theta^\pi$. This enables lower particle budgets through improved particle efficiency, but also introduces the difficulty in controlling this trade-off. In this regard, MCTS-based methods can offer some insight for dealing with this.

Since AlphaZero (Silver et al., 2018), MCTS-based methods also often use a trained policy $\pi_\theta$ (conflated with the name prior distribution) to guide the search in combination with a P-UCT algorithm (Rosin, 2011). Crucially, the initial iterations of the algorithm rely more on $\pi_\theta$, which is interpolated to a greedy policy over the estimated values $Q^\pi$ in later iterations. Grill et al. (2020) show how this causes inferred policies from MCTS to track a regularized objective similarly to Theorem 2.2 (with an added *greediness* parameter) over consecutive iterations. This is relevant to our work, because it links the estimated quantity of our SMC planner to the approach taken by MCTS. The main difference persists in that the iterations of MCTS induce an adaptive regularizer through a proxy for value-accuracy.

Inspired by this, we formulate our proposal distribution through a constrained optimization problem with an adaptive trust-region parameter $\epsilon_\alpha \in \mathbb{R}_{\geq 0}$. At each state $S_t$, the proposal $q$ solves for a locally constrained program,

$$\max_{q \in \mathbb{P}(\mathcal{A}|S_t)} \quad \mathbb{E}_{q(A_t|S_t)} Q_\theta^\pi(S_t, A_t), \qquad (7)$$
$$\text{s.t.,} \quad KL(q(a|S_t)\|\pi_\theta(a|S_t)) \leq \epsilon_\alpha,$$

where $\alpha \in [0, 1]$ is a greediness tolerance level. The actual trust-region $\epsilon_\alpha$ is then sandwiched between the prior $\pi_\theta$ and the greedy policy $\pi^*$ over $Q_\theta^\pi$, such that, $\epsilon_\alpha = \alpha \cdot KL(\pi^*\|\pi_\theta)$. In similar spirit to MCTS, this guarantees that SMC always searches trajectories that interpolate between maximizing $Q_\theta^\pi$ or sticking to the prior $\pi_\theta$. The Lagrangian of this program is similar to Theorem 2.2 with the sum of rewards replaced by $Q_\theta^\pi$ and the $KL$ term scaled by a temperature. Furthermore, Eq. 7 requires $KL(q\|\pi) \leq \epsilon$ at every $S_t \in \mathcal{S}$ instead of in expectation over $p_q(H)$. For solving this program, we use a bisection search using bootstrapped atoms from $\pi_\theta$, which is both general and computationally cheap (see Appendix B.5 for details).

## 3.2. Handling Terminal States with Revived Resampling

Although the control as inference framework from Section 2.1 enables the use of SMC methods for policy inference, it also introduces non-trivial caveats. In particular, the infinite horizon formulation in Eq. 1 can lead to wasting compute when handling terminal states as absorbing states in a forward-only SMC algorithm. Absorbing states result in transitions that loop back to the same state with zero reward, effectively treating this as part of the environment dynamics. In an SMC planner, this can cause some of the $K$ particles to become trapped in these absorbing states. Although resampling should correct for this, reward sparsity and errors in value predictions $V_\theta^\pi \approx V_{soft}^\pi$ can render the weights to become nearly uniform, making these trapped particles indistinguishable from non-trapped ones. This is a problem, because trapped particles do not contribute to gathering information about future values.

Furthermore, the resampling step can also move particles *towards* absorbing states due to rewarding trajectories in previous steps. Consecutive resampling in the SMC planner, however, is then likely to move these particles away from these states again because they no longer accumulate reward. This phenomena is mostly problematic when performing policy inference according to Eq. 6 and is a consequence of the *path degeneracy* problem (Svensson et al., 2015).

To address the trapped particles, we leverage the strategy of MCTS to reset trajectories to earlier states within the search tree. In MCTS, each iteration of the algorithm completely resets the agent to the root state, enabling the accumulation of new trajectory and value information. However, this full reset limits the ability to explore deep inside the search tree, which is a key benefit of SMC with its depth parameter $m$. Therefore, we propose a 'revived resampling' strategy to move particles back to their last non-terminal state. This only requires caching an additional reference state for each particle, assuming that the environment correctly flags these as non-terminal. Resampling is then performed to these reference states instead of the current states (see Appendix C).

## 3.3. Mitigating Path Degeneracy for Policy Inference

A common problem in particle filtering is *path degeneracy*, where most particles collapse to a single ancestor due to resampling. This is illustrated in the top of Figure 1, the grayed-out trajectories highlight the discarded data due to resampling in typical forward-only SMC methods. This loss of ancestor diversity leads to a worse approximation of the distribution over states under our posterior policy (Svensson et al., 2015). We remark that in RL the consequence of this problem is exacerbated since we predominantly care about the root-ancestors. For this reason, SMC planners (Piché et al., 2019; Macfarlane et al., 2024; Lioutas et al., 2023) that perform policy inference for $\hat{q}^*$ using mixtures of point-

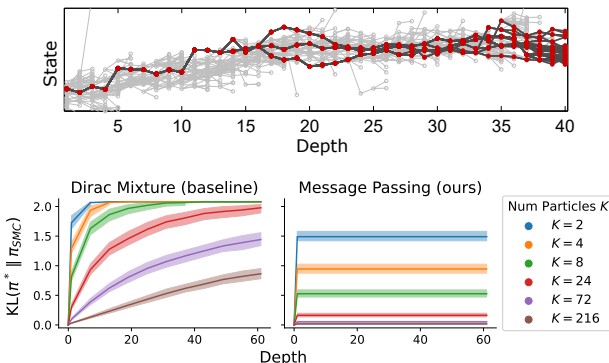

*Figure 1.* Illustration of *path degeneracy* in particle filters (top; adapted from Svensson et al., 2015) and the consequence on divergence from the optimal policy $\pi^*$ (lower is better). With a finite budget of particles and *resampling*, deep search will concentrate the Dirac mixture of remaining ancestors to a single atom (bottom-left). We improve this through approximate message-passing to the ancestors, which does not degenerate the policy (bottom-right).

masses from Eq. 6 show deteriorating approximation quality as depth increases, as shown in Figure 1 (bottom-left).

In contrast, MCTS does not suffer from path degeneracy because it does not discard any data. Policy inference is also typically done in MCTS by tracking a normalized state-action visitation counter that is updated in each iteration of the algorithm (Browne et al., 2012). Importantly, the normalized visit-counts can be used as both a behavior policy and learning target (Silver et al., 2018). Recent work however, has shown that using such a visitation-count policy can degrade performance when using low planning budgets. Instead, Danihelka et al. (2022) show that inferring a regularized policy (Grill et al., 2020) using the value statistics from search for the root state-actions avoids this problem.

However, SMC planners do not track visit-counts or perform backpropagation to accumulate value statistics. Since the importance sampling weights factorize recursively, we can adopt the approach by Moral et al. (2010) to perform an online estimation to $\hat{Q}_{SMC} \approx Q_{soft}^\pi$ inside SMC using Eq. 3, to then construct a policy estimate similarly to Danihelka et al. (2022). At each SMC step $t$, we accumulate a current estimate of $\hat{Q}_{SMC}^{(j)}$ for any root-ancestor $j$ as,

$$\hat{Q}_t^{(j)} = \hat{Q}_{t-1}^{(j)} + \begin{cases} 0, & \mathcal{J}_t^{(j)} = \emptyset, \\ \ln\left(\frac{1}{|\mathcal{J}_t^{(j)}|} \sum_{i \in \mathcal{J}_t^{(j)}} \frac{\widetilde{w}_t^{(i)}}{\widetilde{w}_{t-1}^{(i)}}\right), & \mathcal{J}_t^{(j)} \neq \emptyset, \end{cases}$$

where $\mathcal{J}_t^{(j)} = \{i \in [1,K] \mid J_t^{(i)} = j\}$ and $\hat{Q}_0^{(j)} = 0$. In essence, this expression accumulates an average log-probability for the remaining particles belonging to ancestor $j$. We then use the values $\hat{Q}_{t+m}^{(j)} \approx Q_{soft}^\pi(S_t, A_t^{(j)})$ to infer $\hat{q}^*$ by bootstrapping the atoms proportionally to $\pi_\theta(A_t^{(i)}|S_t) \exp \hat{Q}_{SMC}^{(i)}$. Figure 1 (bottom-right) shows that

this eliminates policy deterioration as a function of depth, reducing the consequence of path degeneracy.

### 3.4. Search-Based Values for Learning Targets

The approximate message passing from the previous subsection mitigates the path degeneracy problem by utilizing all SMC generated data for policy inference. In essence, we are constructing a better estimate of the soft-value functions $V_\theta^\pi \approx V_{soft}^\pi$ (Lawson et al., 2018; Piché et al., 2019) to then estimate $\hat{q}^*$ as given by Eq. 2. The value functions themselves are then trained using some temporal difference (TD) method in an outer learning loop, e.g., using $n$-step returns (Sutton & Barto, 2018) given environment interactions outside of the planner (see also Appendix B.3). So far, most prior work constructs these TD-learning targets by value-bootstrapping from one-step predictions given states in the generated datasets (Piché et al., 2019; Lioutas et al., 2023; Macfarlane et al., 2024). For instance, given a transition $(S_t, A_t, R_t, S_{t+1})$, a 1-step TD target would be computed as $Y_t = R_t + V_\theta^\pi(S_{t+1})$. However, this approach neglects most data generated by the planner by only considering the 1-step value at the next states $S_{t+1}$.

Similarly to recent versions of MuZero (Schrittwieser et al., 2020; Ye et al., 2021; Danihelka et al., 2022), we instead use the values estimated by the search algorithm $\hat{V}_{t+1}$ over $V_\theta^\pi(S_{t+1})$ to compute these outer TD-targets. Not only does this exploit the planner data, $\hat{V}_t$ is also objectively a better value estimator to use for policy improvement (i.e., within our EM-loop). Namely, the predictions by $V_\theta^\pi$ give the value of a previous posterior policy (off-policy), whereas $\hat{V}_t$ is an estimate of the current posterior policy (on-policy). During our testing, we found at low particle budgets for the SMC planner that it is important to control the variance of the inner value estimation. For this reason, we compute $\hat{V}_t$ for value-learning using the Retrace($\lambda$) returns (Munos et al., 2016) instead of the importance-weighted (soft) Monte-Carlo estimate used for the policy. This only requires tracking a secondary statistic along with $\hat{Q}^{(j)}$.

## 4. Experiments

We introduce our new method, *Trust-Region Twisted SMC* (TRT-SMC), a variational SMC method that is tailored for planning in RL. We claim that TRT-SMC implements a stronger approximate *policy improvement* over baseline approaches when used in an expectation-maximization framework (Abdolmaleki et al., 2018; Chan et al., 2022). The contribution of a stronger policy improvement operator can be isolated to 1) enhanced action-selection during training, and 2) improved learning targets (Hamrick et al., 2021). Therefore, we expect our method to yield higher final test returns and steeper learning curves in terms of sample-efficiency (training samples) and runtime efficiency (wall-clock time). We compared our TRT-SMC against the variational SMC method by Macfarlane et al. (2024), and the current strongest Monte-Carlo tree search (MCTS) method, Gumbel AlphaZero by Danihelka et al. (2022).

We performed experiments in the Brax continuous control tasks (Freeman et al., 2021) and Jumanji discrete environments (Bonnet et al., 2024), using the authors' A2C and SAC results as baselines alongside our PPO implementation (Schulman et al., 2017). Although we compare sample-efficiency to the model-free baselines, this is only for reference since we do not account for the additional observed transitions by the planner in the main results. In other words, we are testing the setting where we assume access to a highly accurate simulator for planning purposes (see Appendix D for additional results that also count the simulator samples). Performance is reported as the average offline return over 128 episodes. Unless otherwise stated, all experiments were repeated (retrained) across 30 seeds, with 99% two-sided BCa-bootstrap confidence intervals (Efron, 1987). For more details, see Appendix B.

### 4.1. Main Results

We show the evaluation curves for comparing sample-efficiency in Figure 2, where the planner-based methods used a budget of $N = 16$ transitions. For simplicity, we kept the depth $m$ of the SMC planner uniform to the number of particles $K$, such that $K = m = \sqrt{N}$. Our ablations in the subsection 4.2 also show that keeping $m$ and $K$ somewhat in tandem is ideal for SMC. On each individual environment we observe that *our TRT-SMC shows steeper and higher evaluation curves than the baseline variational SMC method, which is in line with our expectations.*

Additionally, we compare the runtime scaling for additional planning budget in terms of normalized returns on the discrete Jumanji environments in Figure 3. We measured this by aggregating the average returns scaled by their environments' known min-max bounds. We see that with more planning budget that the baseline SMC often starts to approach our TRT-SMC, and that the Gumbel MCTS method scales poorly in wallclock time. Most importantly, these results show that *our TRT-SMC reliably scales in performance with additional budget, runtime, and training samples.*

### 4.2. Ablations

The main results show that our TRT-SMC method can perform better compared to the baseline model-free, variational SMC, and MCTS methods in terms of sample-efficiency and training runtime. However, we also want to asses: *to what extent do each of our separate contributions improve the base method?* Therefore, this section quantifies this across the different environments and parameter settings.

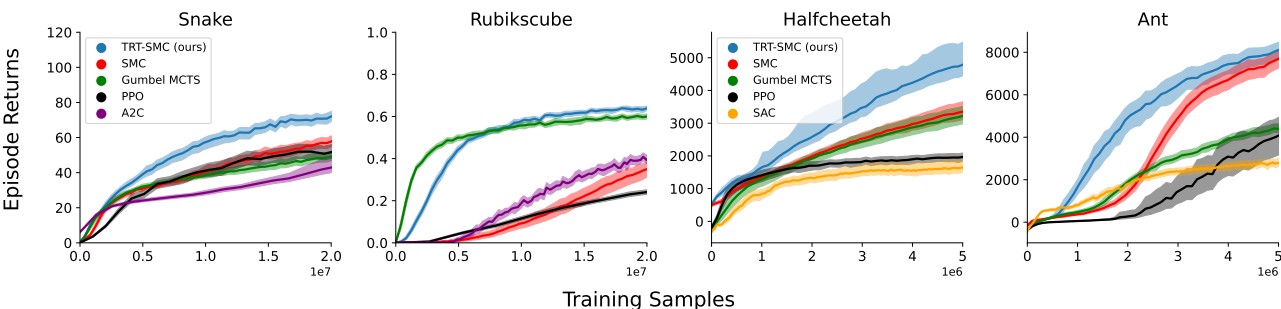

*Figure 2.* Per environment evaluation curves for a planning budget of $N = K \times m = 16$. Shaded regions give 99% two-sided BCa-bootstrap intervals over 30 seeds. This plot shows improved sample-efficiency of our method when a highly accurate simulator is available.

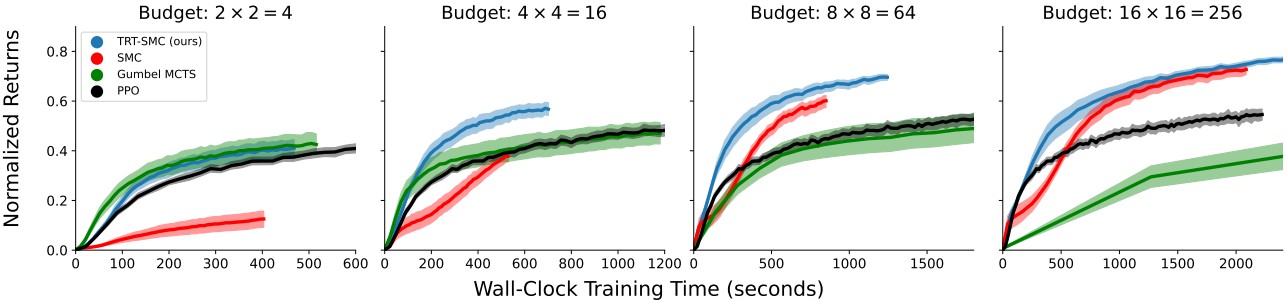

*Figure 3.* Normalized average curves for the discrete environments over increasing planning budgets to compare performance to *runtime*. Shaded regions give 99% two-sided BCa-bootstrap intervals over 2 times 30 seeds. Runtime was estimated by multiplying the training step with an interquartile mean of the runtime-per-step (see Appendix B.2 for details and Appendix D for sample-efficiency).

**Proposal Distributions.** Firstly, we assess the interplay of the proposal distribution with the planning budget through the aggregated final test performance on the Jumanji environments in Figure 4. This compares our TRT-SMC method when using either a twisted proposal distribution with a greediness tolerance of $\alpha = 0.1$, or $\alpha = 0$ (which only uses the prior $\pi_\theta$). Similarly to the main results, at low particle budgets we observe significantly improved performance with the trust-region twisted proposal, but this gap decreases for $\alpha = 0$ at higher particle budgets. It also shows that performance scales favorably when the particle budget and the planner depth are in tandem with eachother. *This performance improvement at low particle budgets matches our aim of increasing particle-efficiency.*

Secondly, we evaluated our TRT-SMC method varying the $\alpha \in [0, 1]$ parameter and uniformly scaling up the budget and depth (as done in Figure 3). We aggregated the normalized final test results over all tested environments as given by the sensitivity plot in Figure 5. Although we show the aggregated results here, we found that this pattern highly depends on the specific environment, we show the individual results in Appendix D. In essence, these results show that *mixing between the prior $\pi_\theta$ and maximizing the predicted state-action value $Q_\theta^\pi$ improves performance compared to completely relying on either of them.*

**Policy Inference.** We compare our method for policy inference to that of Eq. 6 by adopting a similar experiment setup as we used for the proposal ablations and varying these two choices. We report the results for this experiment in terms of their evaluation curves in the right of Figure 4 to also observe learning stability. We find that the final performance of the Dirac policy at $K = 4$ and $\alpha = 0$ shows decreased final performance when the planner depth is increased from $m = 4$ to $m = 16$. As expected, this effect does not occur for our message-passing method, although the proposal twisting seems to diminish this effect also. Most importantly, *our approach for computing $\hat{q}^*$ demonstrates a monotone improvement.*

**Value Targets.** To compare the improvement by the search-based value targets, we tested on the Ant environment for experiment variety. We compared three variations of our TRT-SMC for constructing the *outer* learning targets by linearly interpolating the 1-step $V_\theta$ and the SMC-based value estimate $V_{SMC}$, such that $\hat{V} = \sigma \cdot V_\theta + (1 - \sigma) \cdot V_{SMC}$, where we used $\sigma \in \{0, \frac{1}{2}, 1\}$. We aggregated the final mean performances across different planner depths $m \in \{4, 8\}$, particle budgets $K \in \{4, 8\}$, resampling periods $r \in \{1, 3\}$, and whether to use our revived resampling or not. To reduce moving parts, we did not use a twisted proposal ($\epsilon = 0$). In total, across 30 repetitions, this gave us 480 experiments for which we report final marginal performance in the top of

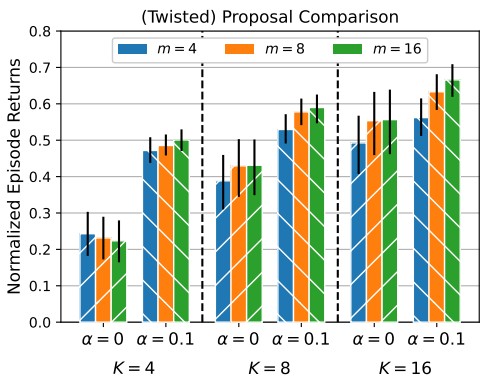
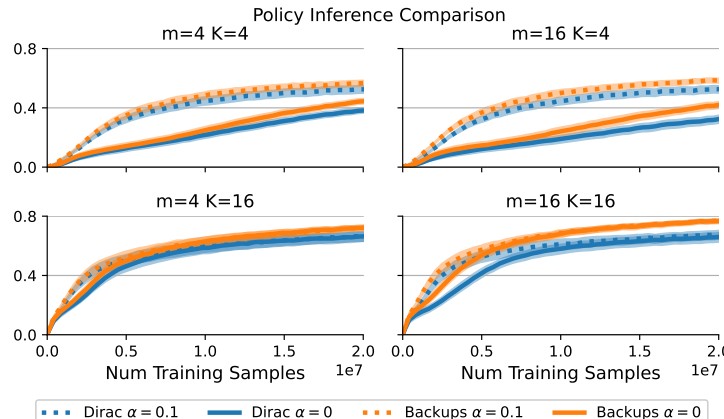

*Figure 4.* Final expected performance (left) and evaluation curves over training (right), for different proposal trust-region levels and policy inference methods. The left barplot only uses our message-passing method (backups) for estimating $\hat{q}^*$, whereas the right plot compares both the Dirac mixture and our method. Both plots show that the constrained proposals $\alpha = 0.1$ improve performance over the prior proposal $\alpha = 0$ at low particle budgets. The right plot also shows that our backup method for $\hat{q}^*$ does not degenerate with deep planning.

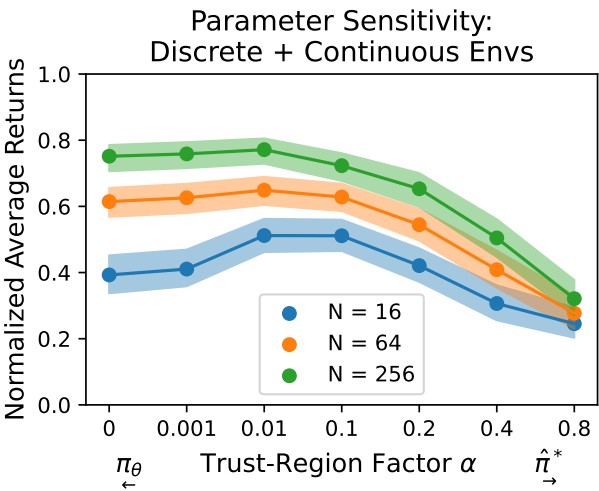

*Figure 5.* Sensitivity plot for the adaptive trust-region parameter $\alpha$ over increasing planning budgets $N = K \cdot m$ (where $K = m$). The $y$-axis indicates the normalized final test performance, aggregated across all tested environments. The pattern shows that at small planning budgets, a proposal distribution which accounts for the predicted $Q_\theta^\pi$ performs marginally better.

Table 1. Although the intervals overlap slightly, *there is a trend that favors using SMC data for the learning targets.*

**Revived Resampling.** We evaluated the effect of using the revived particle resampling in a similar, experiment setup to the value-targets ablations. We tested on Jumanji Snake due to its sparse rewards and high likelihood of encountering terminal states (see Appendix B.1). Interestingly, the marginal test results in the bottom of Table 1 show that *the revived*

*Table 1.* Confidence intervals for the final expected episode returns on Brax Ant for different value-estimation methods (top) and Jumanji Snake for the two resampling strategies (bottom).

| Ablation Value | $\hat{\mu}$ | $\hat{q}_{\alpha/2} - \hat{q}_{1-\alpha/2}$ |
|---|---|---|
| **Value Targets on Ant** | | |
| $V_\theta$ | 6911.8 | $6669.7 - 7146.7$ |
| $\frac{1}{2}V_\theta + \frac{1}{2}V_{SMC}$ | 7214.7 | $6973.9 - 7455.6$ |
| $V_{SMC}$ | **7457.1** | $7200.3 - 7715.3$ |
| **Resampling on Snake** | | |
| Baseline | 44.7 | $43.7 - 45.7$ |
| Revived | 45.7 | $44.5 - 47.7$ |

*resampling does not significantly differ from the baseline.*

## 5. Related Work

The connection of reinforcement learning (RL) to the statistical estimation of a probabilistic graphical model (Levine, 2018) has in recent years proven useful in borrowing tools from Bayesian estimation for optimal policy inference. Although we focus on sequential Monte-Carlo (SMC) methods (Hoffman et al., 2007; Piché et al., 2019), this connection has also been used to exploit stochastic control methods like TD-MPC (Hansen et al., 2024; Theodorou et al., 2010). Similarly, prior work has partially explored some of the modifications that we make to SMC-planners, either in isolation or in different contexts. This paper therefore reinforces the connection between probabilistic inference and RL, but also links it back to recent approaches in Monte-Carlo tree search (MCTS) (Browne et al., 2012; Silver et al., 2018; Schrittwieser et al., 2020; Wang et al., 2024).

As discussed in Section 2.1, our method builds on the varia-

tional SMC approach by Macfarlane et al. (2024). Similarly, they also utilize trust-region methods, but on the neural network parameters during optimization and on the target distribution for SMC. Our setup is more comparable to recent MCTS methods (Danihelka et al., 2022; Wang et al., 2024) since we only impose trust-regions on the proposals and nothing else. In other words, we focus specifically on the *policy improvement* part of the algorithm. Then, Lioutas et al. (2023) also explore a type of 'twisted' proposals, they sample auxiliary action-particles and weight these with heuristic factors $\exp Q_\theta^\pi$ before computing transitions (Stuhlmülller et al., 2015). However, their approach has two issues: they mix the normalization of the auxiliary actions across particles and they do not impose sufficient regularization to trade-off the prior policy and the values.

Finally, our contributions are strongly tied to mitigating path degeneracy in SMC planners, which is a common theme in particle filtering (Chopin & Papaspiliopoulos, 2020). For instance, our estimation of the policy (and values) is similar to the online estimation of any time-separable function described by Moral et al. (2010), which was also motivated by path degeneracy. Although we did not consider it here, there are many promising directions for future work in this area, like using anchor particles (Svensson et al., 2015), adaptive resampling (Naesseth et al., 2019), or Rao-Blackwellisation (Casella & Robert, 1996; Danihelka et al., 2022).

## 6. Conclusion

This paper tailors a particle filter planner for its use within deep reinforcement learning. Specifically, we address default design choices within variational sequential Monte-Carlo that become problematic when applying these methods to perform policy inference. Our contributions take inspiration from recent Monte-Carlo tree search methods to mitigate the path degeneracy problem, make better use of the data generated by the planner, and to improve planning budget utilization. Experiments show that our *Trust-Region Twisted sequential Monte-Carlo* (TRT-SMC) scales favorably to varying planning budgets in terms of runtime and sample-efficiency over the baseline policy improvement methods. We hope that our approach inspires others to leverage more of the tools from both the planning and Bayesian inference literature, to further enhance the sample-efficiency and runtime properties of reinforcement learning algorithms.

## Acknowledgements

JdV and MS are supported by the AI4b.io program, a collaboration between TU Delft and dsm-firmenich, which is fully funded by dsm-firmenich and the RVO (Rijksdienst voor Ondernemend Nederland).

## Impact Statement

This paper advances planning algorithms for use in reinforcement learning. Our improvements specifically enable improved compute scaling, this has the potential to make this field of research more accessible to those with fewer compute resources or improve existing methods at reduced computational cost. This can have diverse societal consequences, none which we feel must be specifically highlighted here.

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

# A. Derivations

We first restate the factorization of the marginal in Eq. 1 from the main text,

$$p_\pi(H_{1:T}) = \prod_{t=1}^{T} \pi(A_t|S_t)p(S_t|S_{t-1}, A_{t-1}),$$

with $p(S_1|A_0, S_0) \triangleq p(S_1)$ being the initial state distribution, $p(S_{t+1}|S_t, A_t)$ is the transition model, and $\pi(A_t|S_t)$ is the policy. We denote the set of admissible policies as $\Pi \triangleq \{\pi|\pi : \mathcal{S} \to \mathbb{P}(\mathcal{A})\}$. We drop subscripts for $H_{1:T}$ if the indexing is clear from the text.

## A.1. Lower-bound

For completeness, we show below that the factorization of $p_\pi(H_{1:T}|\mathcal{O}_{1:T} = 1)$, by Definition 2.1, recovers the lower-bound term for $q^*$ shown in Theorem 2.2. This is done through the well-known decomposition of the log-likelihood on the marginal distribution for the outcome variable $\mathcal{O}_{1:T} \in \{0, 1\}^T$. See Section 2.1 for a description on the meaning of this variable, again we will abbreviate $\mathcal{O} = 1$ simply as $\mathcal{O}$.

**Lemma A.1** (Decomposition log-likelihood, c.f., Ch 9.4 of Bishop (2007))**.**

$$\ln p_\pi(\mathcal{O}) = \underbrace{\mathbb{E}_{p_q(H)} \left[ \sum_{t=1}^{T} R_t - KL(q(a|S_t)\|\pi(a|S_t)) \right]}_{\text{Evidence Lower-Bound}} + \underbrace{KL\left((p_q(H)\|p_\pi(H|\mathcal{O})\right)}_{\text{Evidence Gap}} \tag{8}$$

*Proof.* Assume an importance sampling distribution $q \in \Pi$ for $\pi \in \Pi$ such that it has sufficient support over $H$ and $p_\pi(H) > 0 \implies p_q(H) > 0$ almost everywhere.

$$\begin{aligned}
\ln p_\pi(\mathcal{O}) &= \ln \frac{p_\pi(\mathcal{O}, H)}{p_\pi(H|\mathcal{O})} \\
&= \mathbb{E}_{p_q(H)} \left[ \ln \left( \frac{p_\pi(\mathcal{O}, H)}{p_\pi(H|\mathcal{O})} \frac{p_q(H)}{p_q(H)} \right) \right] \\
&= \mathbb{E}_{p_q(H)} \ln \frac{p_\pi(\mathcal{O}, H)}{p_q(H)} + \mathbb{E}_{p_q(H)} \ln \frac{p_q(H)}{p_\pi(H|\mathcal{O})} \\
&= \mathbb{E}_{p_q(H)} \left[ \ln p(\mathcal{O}|H) + \ln \frac{p_\pi(H)}{p_q(H)} \right] + \mathbb{E}_{p_q(H)} \ln \frac{p_q(H)}{p_\pi(H|\mathcal{O})} \\
&= \mathbb{E}_{p_q(H)} \left[ \ln p(\mathcal{O}|H) - KL(p_q(H)\|\pi(H)) \right] + KL(p_q(H)\|p_\pi(H|\mathcal{O})) \\
&= \mathbb{E}_{p_q(H)} \left[ \sum_t R_t - KL(q(a|S_t)\|\pi(a|S_t)) \right] + KL(p_q(H)\|p_\pi(H|\mathcal{O}))
\end{aligned}$$

where in the last step the transition terms for $p_\pi(H)$ and $p_q(H)$ cancel out. See the work by Levine (2018) for comparison. □

The result from Lemma A.1 shows that the log-likelihood is decomposed into an evidence lower-bound and an evidence gap. This inequality becomes tight when $q$ is simply set to the posterior policy $q^*(H) \equiv p_\pi(H|\mathcal{O})$. Thus, motivating the maximization objective for the lower-bound as given in Theorem 2.2, or equivalently, by minimizing the evidence gap as considered by Levine (2018).

## A.2. Regularized Policy Improvement

The result below gives a brief sketch that the expectation-maximization loop (Neal & Hinton, 1998) generates a sequence of regularized Markov decision processes (MDPs) that eventually converges to a locally optimal policy. This result is a nice consequence of the control-as-inference framework. A more general discussion outside of the expectation-maximization framework can be found in the work by Geist et al. (2019).

**Lemma A.2** (Regularized Policy Improvement). *The solution $q^*$ to the problem,*

$$\max_{q \in \Pi} \mathbb{E}\left[\sum_{t=1}^{T} R_t - KL(q(a|S_t)\|\pi(a|S_t))\right],$$

*guarantees a policy improvement in the unregularized MDP, $\mathbb{E}_{p_{q^*}(H)}[\sum_t R_t] \geq \mathbb{E}_{p_\pi(H)}[\sum_t R_t]$.*

*Proof.* Lemma A.1 shows that the solution $q^*$ is equivalent to the posterior policy distribution $p_\pi(H_{1:T}|O_{1:T})$. This implies that for each state $s \in \mathcal{S}$, we have, $q^*(a|s) \propto \pi(a|s) \exp Q_{soft}^\pi(s,a)$. The exponential over $Q_{soft}^\pi(s,a)$ in the posterior policy $q^*$ interpolates $\pi$ to the greedy policy by shifting probability density to actions with larger expected cumulative reward. Thus, $q^*$ provides a policy improvement over $\pi$. $\qquad\square$

## A.3. Proof of Theorem 2.2

*Proof of Theorem 2.2.* Lemma A.1 shows how the posterior policy coincides with the optimal policy $q^*$ in a regularized Markov decision process (MDP). Then Lemma A.2 describes that this gives a policy improvement in the unregularized MDP. Iterating this process (e.g., an expectation-maximization loop) yields consecutive improvements to the prior $\pi_{(n)} \leftarrow q^*_{(n-1)}$ and guarantees a locally optimal $\pi^*$ in the unregularized MDP as $n \to \infty$, which also implies $KL(q^*_{(n)}\|\pi_{(n-1)}) \to 0$. $\quad\square$

## A.4. Importance sampling weights

For completeness, we give a detailed derivation for the result presented in Corollary 2.3. This derivation differs from the one given by Piché et al. (2019) in Appendix A.4. Our derivation corrects for the fact that we don't need to compute an expectation over the transition function for $\exp V_{soft}^\pi(S_t)$ in the denominator. However, this is only a practical difference (i.e., how the algorithm is implemented) to justify our calculation.

**Corollary A.3** (Restated Corollary 2.3). *Assuming access to the transition model $p(S_{t+1}|S_t, A_t)$, we obtain the importance sampling weights for $p_\pi(H_{1:t}|\mathcal{O}_{1:T})/p_q(H_{1:t})$,*

$$w_t = w_{t-1} \cdot \frac{\pi(A_t|S_t)}{q(A_t|S_t)} \exp(R_t) \frac{\mathbb{E}[\exp V_{soft}^\pi(S_{t+1})]}{\exp V_{soft}^\pi(S_t)},$$

*Proof.* For any statistic $f(\cdot)$, we have,

$$\mathbb{E}_{p_\pi(H_{1:t}|\mathcal{O}_{1:T})}f(H_{1:t}) = \mathbb{E}_{p_q(H_{1:t})}\left[w_t \cdot f(H_{1:t})\right]$$

$$= \mathbb{E}_{p_q(H_{1:t})}\left[\frac{p_\pi(H_{1:t}|\mathcal{O}_{1:T})}{p_q(H_{1:t})}f(H_{1:t})\right]$$

$$= \mathbb{E}_{p_q(H_{1:t})}\left[\frac{p_\pi(S_t, A_t|H_{<t}, \mathcal{O}_{1:T})p_\pi(H_{<t}|\mathcal{O}_{1:T})}{p_q(S_t, A_t|H_{<t})p_q(H_{<t})}f(H_{1:t})\right]$$

$$= \mathbb{E}_{p_q(H_{1:t})}\left[w_{t-1} \cdot \frac{p_\pi(S_t, A_t|S_{t-1}, A_{t-1}, \mathcal{O}_{t:T})}{p_q(S_t, A_t|S_{t-1}, A_{t-1})} \cdot f(H_{1:t})\right]$$

where the last step follows from the Markov property. Then, we get,

$$\frac{p_\pi(S_t, A_t|S_{t-1}, A_{t-1}, \mathcal{O}_{t:T})}{p_q(S_t, A_t|S_{t-1}, A_{t-1})} = \frac{p_\pi(A_t|S_t, \mathcal{O}_{t:T})p(S_t|S_{t-1}, A_{t-1})}{q(A_t|S_t)p(S_t|S_{t-1}, A_{t-1})} = \frac{p_\pi(A_t|S_t, \mathcal{O}_{t:T})}{q(A_t|S_t)}$$

$$= \frac{\pi(A_t|S_t)}{q(A_t|S_t)} \frac{p_\pi(\mathcal{O}_{t:T}|S_t, A_t)}{p_\pi(\mathcal{O}_{t:T}|S_t)} = \frac{\pi(A_t|S_t)}{q(A_t|S_t)} \frac{\exp Q_{soft}^\pi(S_t, A_t)}{\exp V_{soft}^\pi(S_t)}$$

$$= \frac{\pi(A_t|S_t)}{q(A_t|S_t)} \exp(R_t) \frac{\mathbb{E}[\exp V_{soft}^\pi(S_{t+1})]}{\exp V_{soft}^\pi(S_t)}.$$

$\square$

# B. Experiment Details

Our code can be found at `https://github.com/joeryjoery/trtpi`.

## B.1. Environments

We used the Jumanji 1.0.1 implementations of the Snake-v1 and Rubikscube-partly-scrambled-v0 environments (Bonnet et al., 2024), code is available at `https://github.com/instadeepai/jumanji`. For the Brax 0.10.5 implementation we used the Ant and Halfcheetah environments using the 'spring' backend, code is available at `https://github.com/google/brax`.

Snake environment details:

- The observation is a 12x12 image with 5 channels that indicate the position of the fruit and positional features of the snake, it also gives the integer number of steps taken which we encode as a bit-vector for the neural network.

- The action space is a choice over 4 integers that indicate moving the snake: up, down, left, or right.

- The reward is zero everywhere, except for a +1 when the fruit is picked up.

- The agent terminates after 4000 steps or when colliding with itself.

Rubikscube environment details:

- The observation is a 3D integer tensor of shape $6 \times 3 \times 3$ with values in $[0, 1, 2, 3, 4, 5]$ indicating the color. It also gives the integer number of steps taken which we encode as a bit-vector for the neural network.

- The action space is a 3 dimensional integer array to choose the face to turn, depth of the turn, and direction of the turn. For a 3x3 cube this action-space induced 18 combinations.

- The reward is zero everywhere, except for a +1 when the puzzle is solved.

- The agent terminates after 20 steps or when solving the puzzle.

- We used the partly-scrambled version of this environment, which means that the solution is at most 7 actions removed from any of the starting states.

Brax environment details:

- The observations are continuous values with 27 dimensions for Ant and 18 dimensions for Halfcheetah.

- The action space is a bounded continuous vector between $[-1, 1]^D$, with $D = 8$ for Ant and $D = 6$ for Halfcheetah.

- The reward is a dense, but involved formula that penalizes energy expenditure (norm of the action) and rewards the agent for moving in space (in terms of spatial coordinates).

- The agent terminates after 4000 steps or when ending up in a unhealthy joint-configuration.

## B.2. Hardware Requirements

All experiments were run on a GPU cluster with a mix of NVIDIA GeForce RTX 2080 TI 11GB, Tesla V100-SXM2 32GB, NVIDIA A40 48GB, and A100 80GB GPU cards (Delft AI Cluster (DAIC), 2024; Delft High Performance Computing Centre , DHPC). Each run (random seed/ repetition) required only a few CPU cores (2 logical cores) with a low memory budget (e.g., 4GB). For our most expensive singular experiments we found that we needed about 6GB of VRAM at most, and that the replay buffer size is the most important parameter in this regard. Roughly speaking, we found that the SMC based agents all completed both training and evaluation under 30 minutes on Snake with a budget of $K = 8$ particles and a depth of $m = 8$, the Gumbel MCTS required 3 hours for a budget of $N = 64$.

**Training Time Estimation.** To estimate the training runtime in seconds (in Figure 3), we used an estimator of the the runtime-per-step and multiplied this by the current training iteration to obtain a cumulative estimate. For each training configuration, we measured the runtime-per-step and computed an interquartile mean over 1) the random seeds for relative wallcock time and 2) the training iterations themselves. This estimator should more robustly deal with the variations in hardware, the compute clusters' background load, and XLA dependent compilation. Of course, estimating runtime is strongly limited to the hardware and software implementation, and our results should only hint towards a trend of improved scaling to parallel compute for the planner algorithms.

### B.3. Hyperparameters, Model Training, and Software Versioning

The hyperparameters for our experiments are summarized in the following tables:

- Table 3: Shared parameters across experiments.

- Table 4: PPO-specific parameters.

- Table 5: MCTS-specific parameters.

- Table 6: Shared SMC parameters.

- Table 7: Parameters for our extended agent.

We underline all default values in bold, all other parameter values indicated in the sets were run in an exhaustive grid for the ablations. The ablation results then report marginal performance over configurations and over seeds (repetitions). These experimental design decisions closely follow the suggestions laid out in the work by Patterson et al. (2024).

Despite conflating all model parameters into one joint set $\theta$, we used separate neural network parameters for the policy, state, and state-action value models. Given the current dataset (replay buffer) within the training loop $\mathcal{D}_{(n)}$, the loss is a simple empirical cross-entropy of collective terms,

$$\mathcal{L}(\theta) = \mathbb{E}_{(S_t, A_t, \hat{q}_t, \hat{V}_t) \sim \mathcal{D}_{(n)}} \left[ \frac{c_v}{2}(\hat{V}_t - V_\theta^\pi(S_t))^2 + \frac{c_v}{2}(\hat{V}_t - Q_\theta^\pi(S_t, A_t))^2 - c_\pi \mathbb{E}_{a \sim \hat{q}_t} \ln \pi_\theta(a|S_t) - c_{ent} \mathcal{H}[\pi_\theta(a|S_t)] \right] \tag{9}$$

where $\hat{q}_t$ and $\hat{V}_t$ are estimated targets for the policy and value respectively (see main text), and $\mathcal{H}[\pi_\theta] = -\mathbb{E}_{\pi_\theta} \ln \pi_\theta$ is an entropy penalty for the policy. We approximated $\mathcal{L}$ with stochastic gradient descent (see the hyperparameter tables), where we always used the AdamW optimizer (Loshchilov & Hutter, 2019) with an $l_2$ penalty of $10^{-6}$ and a learning rate of $3 \cdot 10^{-3}$. Gradients were clipped using two methods, in order: a max absolute value of 10 and a global norm limit of 10.

The replay buffer was implemented as a uniform circular buffer with its size calculated as:

$$\text{max-age of data} \times \text{number of parallel environments} \times \text{number of unroll steps.}$$

The max-age of data was tuned to fit into reasonable GPU memory.

For the A2C baseline, hyperparameters are detailed in Bonnet et al. (2024), and for the SAC baseline, refer to Freeman et al. (2021). Additionally, Table 2 provides version information for key software packages, this also directs towards default hyperparameters of baselines not listed here. We implemented everything based on Jax 0.4.30 in Python 3.12.4.

### B.4. Neural Network Architectures

Neural network designs were largely adapted from the A2C reference (Bonnet et al., 2024) and Macfarlane et al. (2024). Minor modifications were made to handle the heterogeneity in environment action spaces. Notably, we standardized network architectures across environments wherever feasible, adjusting input embedding and output construction as needed. The specific configurations are listed below.

**Brax Environments**

- 2-layer MLP with 256 nodes per layer.

*Table 2.* Software module versioning that we used for our experiments (also includes default parameter settings).

| Package | Version |
|---------|---------|
| brax | 0.10.5 |
| optax | 0.2.3 |
| flashbax | 0.1.2 |
| rlax | 0.1.6 |
| mctx | 0.0.5 |
| flax | 0.8.4 |
| jumanji | 1.0.1 |

- Leaky-ReLU activations followed by LayerNorm.

- Outputs parameterized a diagonal multivariate Gaussian squashed via Tanh, as in Haarnoja et al. (2018).

**Jumanji RubiksCube Environment**

- 2-layer MLP with 256 nodes per layer (same as Brax).

- We used a flat representation for the 3-dimensional categorical action space (logits over all item-combinations). In contrast: the A2C baseline used a structured representation with three separate categorical outputs (logits per item).

**Jumanji Snake Environment**

- 2-layer MLP with 128 nodes per layer and Leaky-ReLU activations, followed by LayerNorm for the main module.

- Based on Bonnet et al. (2024), the input image is embedded using a single 3x3 convolutional layer with:

    - 3 channels.
    - Leaky-ReLU activation.
    - No LayerNorm.

- The resulting embedding was flattened before being passed to the main MLP module.

*Table 3.* Shared experiment hyperparameters.

| Name | Symbol | Value Jumanji | Value Brax |
|---|---|---|---|
| SGD Minibatch size | | 256 | 256 |
| SGD update steps | | 100 | 64 |
| Unroll length (nr. steps in environment) | | 64 | 64 |
| Batch-Size (nr. parallel environments) | | 128 | 64 |
| (outer-loop) TD-Lambda | $\lambda$ | 0.95 | 0.9 |
| (outer-loop) Discount | $\gamma$ | 0.997 | 0.99 |
| Value Loss Scale | $c_v$ | 0.5 | 0.5 |
| Policy Loss Scale | $c_\pi$ | 1.0 | 1.0 |
| Entropy Loss Scale | $c_{ent}$ | 0.1 | 0.0003 |

*Table 4.* Proximal Policy Optimization hyperparameters. We did not use advantage-normalization computed the policy entropy exactly.

| Name | Symbol | Value Jumanji | Value Brax |
|---|---|---|---|
| Policy-Ratio clipping | $\epsilon$ | 0.3 | 0.3 |
| Value Loss Scale | $c_v$ | 1.0 | 0.5 |
| Policy Loss Scale | $c_\pi$ | 1.0 | 1.0 |
| Entropy Loss Scale | $c_{ent}$ | 0.1 | 0.0003 |

*Table 5.* Gumbel Monte-Carlo tree search experiment hyperparameters (Danihelka et al., 2022).

| Name | Symbol | Value Jumanji | Value Brax |
|---|---|---|---|
| Replay Buffer max-age | | 64 | 64 |
| Nr. bootstrap atoms $\pi$ | $B$ | 30 | 30 |
| Search budget | $N$ | $\{16, 64\}$ | $\{16, 64\}$ |
| Max depth | | 16 | 16 |
| Max breadth | | 16 | 16 |

*Table 6.* Shared Sequential Monte-Carlo hyperparameters (ours and Macfarlane et al., 2024). Bold values indicate those used in the main results, with the remaining values in the set being explored in the ablations.

| Name | Symbol | Value Jumanji | Value Brax |
|---|---|---|---|
| Replay Buffer max-age | | 64 | 64 |
| Planner Depth | $m$ | $\{\mathbf{4}, \mathbf{8}, 16\}$ | $\{\mathbf{4}, \mathbf{8}\}$ |
| Number of particles | $K$ | $\{\mathbf{4}, \mathbf{8}, 16\}$ | $\{\mathbf{4}, \mathbf{8}\}$ |
| Resampling period | $r$ | $\{1, \mathbf{3}\}$ | $\{1, \mathbf{3}\}$ |
| Target temperature (env. reward scale) | $T$ | $\{1.0, \mathbf{0.1}\}$ | $\{1.0, \mathbf{0.1}\}$ |
| Nr. bootstrap atoms $\pi$ | $B$ | 30 | 30 |

*Table 7.* Trust-Region Twisted Sequential Monte-Carlo hyperparameters (i.e., ours only). The underlined values recover the base SMC.

| Name | Symbol | Value Jumanji | Value Brax |
|---|---|---|---|
| (inner-loop) Retrace($\lambda$) | $\lambda_{SMC}$ | 0.95 | 0.9 |
| (inner-loop) Discount | $\gamma_{SMC}$ | 0.997 | 0.99 |
| (outer-loop) Value mixing $\hat{V}_t$ | $\sigma$ | 0.5 (0.0) | $\{\underline{0.0}, \mathbf{0.5}, 1.0\}$ |
| Estimation $\hat{q}^*$ | | $\{\underline{\text{Dirac}}, \mathbf{\text{Message-Passing}}\}$ | $\{\underline{\text{Dirac}}, \mathbf{\text{Message-Passing}}\}$ |
| Revived resampling | | $\{\underline{\text{False}}, \mathbf{\text{True}}\}$ | $\{\underline{\text{False}}, \mathbf{\text{True}}\}$ |
| Proposal (adaptive) Trust-Region | $\epsilon_\alpha$ | $\{\underline{0.0}, \mathbf{0.1}, 0.3\}$ | $\{\underline{0.0}, \mathbf{0.1}, 0.3\}$ |

### B.5. Details on Constrained Proposals

We solve the constrained program in Eq. 7 in the SMC planner in Algorithm 1 for each individual state-particle $S_t^{(i)}$. To deal with general action-spaces (e.g., continuous), we sample $B$ atoms from the prior policy $\pi_\theta$ for uniform bootstrapping. As stated in Theorem 2.2 and accompanying text, the Lagrangian of this program can be used to define a Boltzmann policy,

$$L(q, \beta^{-1}, \eta) = \mathbb{E}_q Q_\theta^\pi + (\epsilon_\alpha - \beta^{-1} KL(q\|\pi_\theta)) + (1 - \eta), \tag{10}$$

where taking the partial derivatives and setting them to zero gives $q^* \propto \pi_\theta \exp \beta Q_\theta^\pi$, which is analytically normalizable (see Grill et al. (2020) for comparison). Given this distribution $q^*$, we found that the following minimization problem was the most numerically stable in finding the optimal temperature parameter $\beta^{-1}$,

$$\min_{\beta^{-1}} \| \epsilon_\alpha - \mathbb{E}_{q^*}[\ln q^*(a|S) - \ln \pi_\theta(a|S)] \|_2^2, \tag{11}$$

which we solve with a bisection search. In combination with bootstrapping from $\pi_\theta$, the above $\ln \pi_\theta(a|S)$ is essentially an entropy constraint on $q^*$. As stated in Subsection 3.1, we set $\epsilon_\alpha$ adaptively based on some greediness tolerance $\alpha \in [0, 1]$.

### B.6. Details on Figure 1

We adapted the top-figure with the colored and grayed out trajectories from Figure 2 of Svensson et al. (2015) to show the discarded data in a naive forward-only sequential Monte-Carlo (SMC) planner. We generated the two bottom figures using our own SMC planner implementation. The KL-divergence was evaluated by: running SMC at timestep 0 on a dummy environment, extracting the sampled policy as by Eq. 6 or from Section 3.3, projecting the sampled logits back to their original action-space, and calculating the KL-divergence of the optimal stochastic policy to the canonical (non-bootstrapped) logits from SMC. Most importantly, this environment had discrete actions and zero reward everywhere, making the dynamics function like an absorbing state. For this reason, the optimal (stochastic) policy is uniformly random with a value $V^\pi$ of zero everywhere. The bottom-left of Figure 1, thus, visualizes the consequence of recursive bootstrapping, which degenerates the policy in terms of KL-divergence from the optimal policy. Our method does not incur this in the bottom-right aside from the effect caused by the particle budget.

## C. Pseudocode

We give a simplified overview of our method in Algorithm 2, which is an extended version of the one from the main paper (Algorithm 1). The pseudocode documents our specific contributions in comparison to the base SMC planner.

Note that the implementation for the online tracking of ancestor values, following Moral et al. (2010) is very similar to the eligibility trace known in reinforcement learning (Sutton & Barto, 2018; Seijen & Sutton, 2014). Fundamentally, this can be considered as initializing the eligibilities of the ancestor particles to one, and continuously decaying these eligibilities while accumulating value updates (i.e., without updating the ancestor-eligibility).

---

**Algorithm 2** Bootstrapped Particle Filter for RL (Our TRT-SMC Pseudocode based on Algorithm 1).

---

**Require:** $K$ (number of particles), $m$ (depth), $r$ (resampling period), $\alpha$ (proposal greediness)

1: Initialize:
- Ancestor identifier $\{J_1^{(i)} = i\}_{i=1}^K$
- States $\{S_1^{(i)} \sim p(S_1)\}_{i=1}^K$
- Reference States $\{\widetilde{S}_1^{(i)} \leftarrow S_1^{(i)}\}_{i=1}^K$          // Revived resampling: track last non-terminal states
- Weights $\{\widetilde{w}_0^{(i)} = 1\}_{i=1}^K$
- Ancestor Log-Probabilities $\{\hat{Q}_0^{(i)} = 0\}_{i=1}^K$          // Policy Inference

2: **for** $t = 1$ to $m$ **do**

    // TRT-SMC: Create a set of Trust-Region twisted proposal distributions

3:     $\{q_t^{(i)} \mid$ where $q_t^{(i)}$ solves Eq. 7 for a given $\alpha, S_t^{(i)}\}_{i=1}^K$

    // Default SMC: Update particles

4:     $\{A_t^{(i)} \sim q_t^{(i)}(A_t|S_t^{(i)})\}_{i=1}^K$

5:     $\{S_{t+1}^{(i)} \sim p(S_{t+1}|S_t^{(i)}, A_t^{(i)})\}_{i=1}^K$

6:     $\{\widetilde{w}_t^{(i)} = \widetilde{w}_{t-1}^{(i)} \frac{\pi(A_t^{(i)}|S_t^{(i)})}{q(A_t^{(i)}|S_t^{(i)})} e^{R_t^{(i)}} \frac{\mathbb{E} \exp V_\theta^\pi(S_{t+1}^{(i)})}{\exp V_\theta^\pi(S_t^{(i)})}\}_{i=1}^K$

    // Revived resampling: Track the last non-terminal states

7:     $\{\widetilde{S}_{t+1}^{(i)} \leftarrow \begin{cases} \widetilde{S}_t^{(i)}, & \text{If } S_{t+1}^{(i)} \text{ is terminal,} \\ S_{t+1}^{(i)}, & \text{Otherwise,} \end{cases} \}_{i=1}^K$

    // Policy inference & Value Estimation: online accumulation of ancestor statistics

8:     $\{\mathcal{J}^{(j)} \leftarrow \{i \in [1, K] \mid J_t^{(i)} = j\}\}_{j=1}^K$

9:     $\hat{Q}_t^{(j)} = \hat{Q}_{t-1}^{(j)} + \begin{cases} 0, & \mathcal{J}_t^{(j)} = \emptyset, \\ \ln\left(\frac{1}{|\mathcal{J}_t^{(j)}|} \sum_{i \in \mathcal{J}_t^{(j)}} \frac{\widetilde{w}_t^{(i)}}{\widetilde{w}_{t-1}^{(i)}}\right), & \mathcal{J}_t^{(j)} \neq \emptyset, \end{cases}$

    // Default SMC: Bootstrap (periodically) through resampling

10:    **if** $t \bmod r == 0$ **then**

11:       Normalized probability vector: $\overline{w}_t[i] = \frac{\widetilde{w}_t^{(i)}}{\sum_{j=1}^K \widetilde{w}_t^{(j)}}$

12:       $\{J_t^{(i)} \sim \text{Categorical}(\overline{w}_t)\}_{i=1}^K$

      // Revived resampling: resample particles by their reference states, then reset the references.

13:       $\{(S_{t+1}^{(i)}, A_t^{(i)})\}_{i=1}^K \leftarrow \{(\widetilde{S}_{t+1}^{(J_t^{(i)})}, A_t^{(J_t^{(i)})})\}_{i=1}^K$

14:       $\{\widetilde{S}_{t+1}^{(i)} \leftarrow S_{t+1}^{(i)}\}_{i=1}^K$

15:       $\{\widetilde{w}_t^{(i)} = 1\}_{i=1}^K$

16:    **end if**

17: **end for**

    // Return the estimated values for performing policy inference

18: **return** $\{\hat{Q}_m^{(j)}\}_{i=1}^K$

---

---

**Algorithm 3** Outer EM-loop for Approximate Policy Iteration

---

**Require:** Initial iterate $\theta_{(1)}$ for neural networks, replay buffer $\mathcal{D}_{(1)}$.

1: **for** $n = 1$ to $N$ **do**
2:     $S_1 \sim p(S_1)$
3:     **for** $t = 1$ to $T$ **do**
        // Inner-loop; Model-Predictive Control (Algorithm 1)
4:         $\{J_{t:t+m}^{(i)}, H_{t:t+m}^{(i)}, \widetilde{w}_{t:t+m}^{(i)}\}_{i=1}^{K} \sim \text{SMC}(S_t; \pi_{\theta_{(n)}}, V_{\theta_{(n)}}^{\pi})$
5:         Estimate $\hat{q}_t^*$, a policy using SMC-output (e.g., Eq 6).
6:         Estimate $\hat{V}_t'$, a (inner) value using SMC-output (e.g., $V_{\theta_{(n)}}^{\pi}(S_t)$).
7:         Sample action from search policy $A_t \sim \hat{q}_t^*$
        // Environment step and data collection
8:         $S_{t+1} \sim p_{env}(\cdot | S_t, A_t), R_t \sim R(S_t, A_t)$
9:         Append $(S_t, A_t, R_t, \hat{q}_t^*, \hat{V}_t')$ to buffer $\mathcal{D}_{(n)}$
10:    **end for**
        // Outer loop; learning through SGD
11:    Compute (outer) value estimators $\hat{V}_t$ using rewards $R_t$ and (inner) values $\hat{V}_t'$ from $\mathcal{D}_{(n)}$ (e.g., truncated TD($\lambda$))
12:    Update $\theta_{(n+1)}$ with SGD on $\mathcal{L}(\theta_{(n)}; \mathcal{D}_{(1:n)})$ (Equation 9)
13:    (Optionally: Circularly wrap $\mathcal{D}_{(n+1)}$ from $\mathcal{D}_{(n)}$)
14: **end for**
15: **return** $\pi_{\theta_{(N+1)}}$

---

# D. Supplementary Results

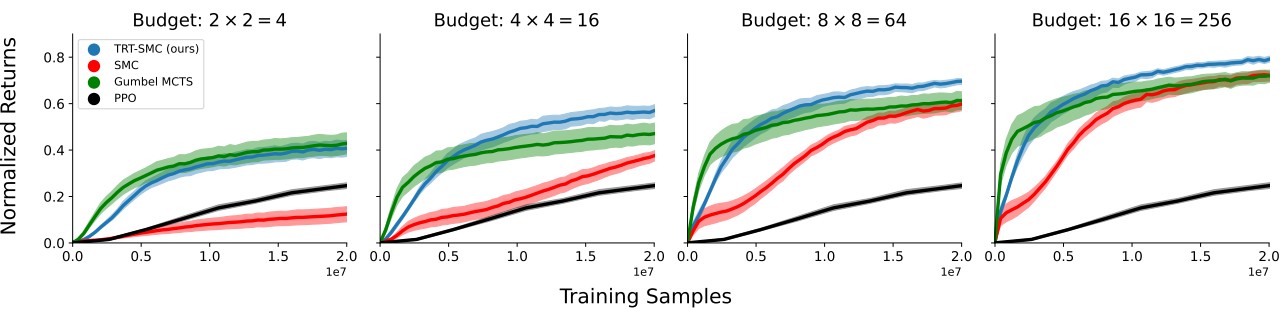

*Figure 6.* Normalized expected evaluation curves for the discrete environments over increasing planning budgets. These budgets were calculated as $N = K \times m$, where $K = m$, to keep the number of transitions uniform between the MCTS and SMC methods. Shaded regions give BCa $\alpha = 0.01$ intervals over 2 times 30 seeds. See also Figure 3 in the main text for a similar comparison to runtime scaling.

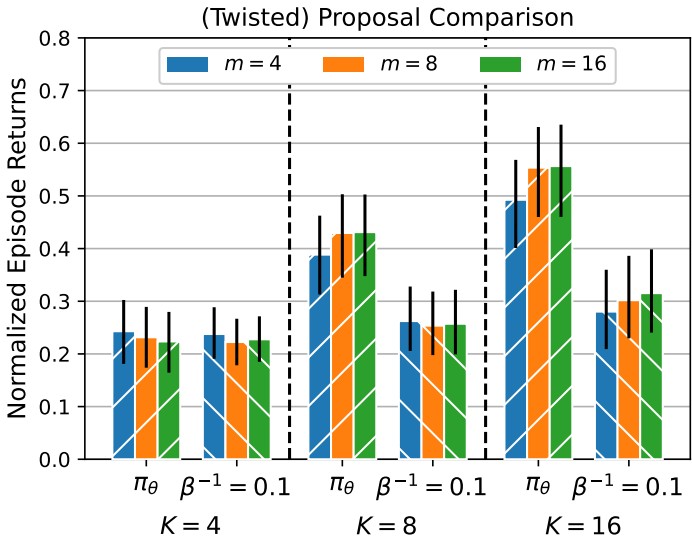

*Figure 7.* Final expected performance for the prior $\pi_\theta$ and the regularized proposal with a static temperature of $\beta^{-1} = 0.1$. This plot is identical to the left barplot in Figure 4 and shows that taking values into account for the proposal doesn't directly translate to improved performance (i.e., the approach taken by Lioutas et al., 2023). We found it essential for performance that the temperature $\beta^{-1}$ must be set adequately, which we achieved with the adaptive trust-region method.

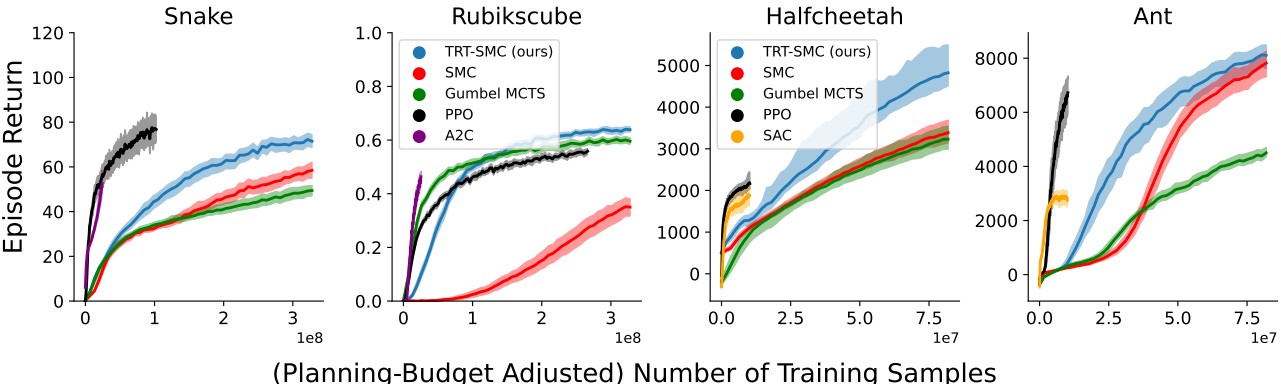

(Planning-Budget Adjusted) Number of Training Samples

*Figure 8.* Main results from Figure 2 with the adjusted $x$-axis to account for additional true environment samples during planning. In terms of sample-efficiency this gives a stark contrast to the result of the main paper, and shows that the model-free methods are more sample efficient. However, methods that utilize planning can always utilize an accurate simulator to skew the $x$-axis similarly to the result of Figure 2.

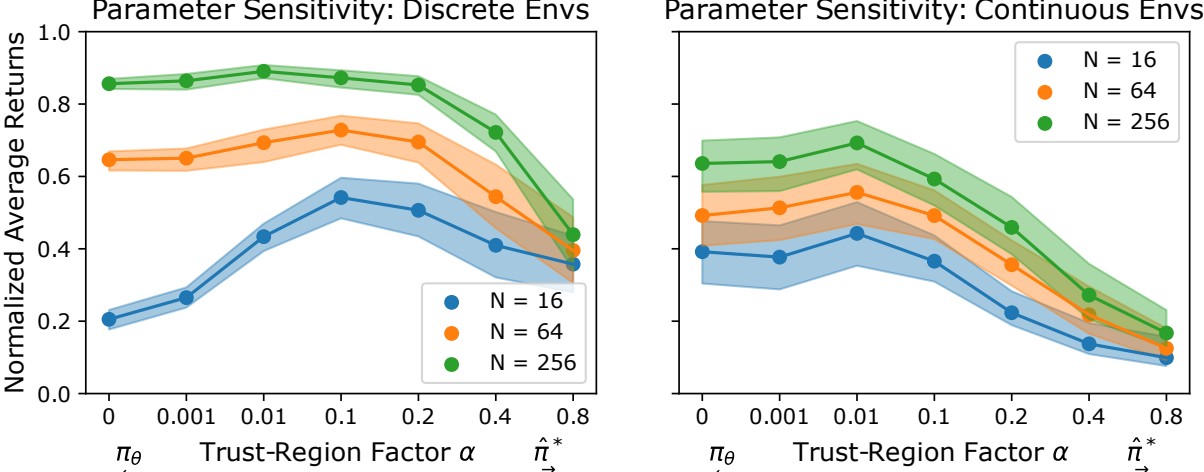

*Figure 9.* Sensitivity plot for the adaptive trust-region parameter $\alpha$ split for the discrete (Jumanji) and continuous (Brax) environments. This splits up the results shown in Figure 5 from the main paper.

