# OpenReview forum: "Trust-Region Twisted Policy Improvement"
_ICML.cc/2025/Conference — ICML 2025 poster_

### Official Review · Reviewer_Pz2A · 2025-03-03

**Overall Recommendation:** 3

**Summary:**

The authors propose several improvements to SMC:

1. Addressing path degeneracy by performing an online estimate of $Q_{SMC}$.
2. Using the $V$ values obtained by the search algorithm.
3. Reviving trapped particles.
4. Changing the proposal policy to be in the trust region.

The authors provide experiments to show their improvement over the original SMC and a recent variation of MCTS.

**Claims And Evidence:**

The authors propose 4 changes to SMC, it would make sense to test the merit of each separately in a clear and simple figure.
I found the figures to be complicated and I don't think such a simple test was done on all 4 improvements.

**Essential References Not Discussed:**

Not that I know of.

**Experimental Designs Or Analyses:**

No. The plots in Figure 2 look reasonable to me.

**Methods And Evaluation Criteria:**

Yes.

**Other Comments Or Suggestions:**

None.

**Other Strengths And Weaknesses:**

Even though the paper has mostly technical improvements I think it provides a solid advancement for SMCs and can be beneficial to the community.

In my view, the paper has two weaknesses:

1. Clarity - The paper is full with details and it's hard to follow. I'd like to be more specific, but I think that it's the entire paper - it's just complicated. What I would think would immensely are diagrams showing the learn process and the proposed improvements.

Also the plots could be simpler:

In Figure 1 what is the y-axis of the message passing plot? is it really figure-worthy if its a constant value (maybe it is possible to  calculated it directly?)

Figure 3 is in the main paper but only referenced in the appendix, Figure 5 is in the appendix but referenced in the main.

2. There are several claims like X is problematic in SMC and we solve it with Y. But how much X is problematic is not always clear or shown. Also the improvement with the solution is not always shown and some simple ablation study is needed to just compare the final result each improvement with and without. I understand it's not always easy, but that seems to be the main contribution of the paper.

**Questions For Authors:**

1. Can you show that each improvement you proposed on its own improved the training curve, and also show that it address the issue you've raised?

**Relation To Broader Scientific Literature:**

It is an improvement to SMC.

**Theoretical Claims:**

No - Thm 2.2 is taken from another paper, Corollary  2.3 has no full proof but makes sense. No other theoretical proof is given.

---

> ### Author Rebuttal · Authors · 2025-04-01
>
> We thank the reviewer for their effort and accurate summary of our paper, we appreciate that you recognize our work as a solid advancement to SMC-planners for RL and the positive assessment.
>
> We address your points below.
>
> ---
>
> **Minor comments:**
>
>  - We included the derivation for Corollary 2.3 in the appendix A.4, but forgot to reference this. We will update this.
>  - The reference to Fig.5 vs Fig.3 in the main text and appendix is a mistake, this should be swapped. Good catch!
>
> ---
>
> **Figure 1:**
>
> In Fig.1 we show that running the baseline SMC planner for a longer horizon (depth) causes the distribution over actions $\pi_{SMC}$ at the start to diverge from their optimal distribution $\pi^*$. This is indicated in the y-label with $KL(\pi^* \Vert \pi_{SMC})$. We believe it's important to show the pattern that our method doesn't worsen as a function of depth. The complete details on how we generated this figure are in Appendix B.6, we will add a reference for this in the figure 1 caption.
>
> This pattern is a problem in the baseline because it increases variance of the policy estimation which cascades into exploration and learning (as also noted by Reviewer bnoX). We tested our modification in the ablation-study in figure 4 on the right. We tested this over multiple parameter settings due to hyperparameters interacting with one-another. This follows principles laid out by Patterson et al. (2024). See also our answer below on the ablations.
>
> For clarity, we will copy the ylabel in Fig.1 from the left-bottom also to the plot in the right-bottom.
>
> ---
>
> **Learning diagram:**
>
> We prepared pseudocode for the outer loop to complement our discussion on approximate policy iteration in the background and the inner-loop pseudocode in the appendix (see also our response to reviewer SRnR on how we intend to revise this). The pseudocode can be found here: [link to EM-pseudocode](https://www.dropbox.com/scl/fi/ktxtkcuq00syv0k7i7vew/outer-loop-pseudocode.png?rlkey=dwxroqk5xrl629v7a4oesye9c&st=07wviyru&dl=0)), we will put this either in Appendix C or in the main text depending on whether we have space for additional ablation results.
>
> **Ablations + Answer to Question:**
>
> We tested out all the improvements separately to the best of our abilities and resources, across varying environments and marginalizing over many parameter settings. This is mostly shown in figure 4 and table 1. Because there were so many hyperparameters involved, the details of the ablations were put in the appendix B.3. Furthermore, we simply had to make sacrifices in what to test within this immense combinatorial space of hyperparameters.
>
> We recognize that the details of each ablation study got a bit buried in section 5 when trying to connect this back to section 3, where we described the improvements.
>
> **Q:** As a minor revision, we will annotate each section in the experiments. Would this help?
>
> For example, the contribution in section 3.2 is tested in the experiments in line 434 (col 1), we propose to 1) reference this part in section 3.2 and 2) cross-reference back to section 3.2 in the paragraph-heading -- e.g., **Revived Resampling (§3.2)**.'
>
> In summary,
>  - §3.1 improves sampling inside SMC, this is studied in figure 4.
>  - §3.2 prevents particles from getting trapped in absorbing states, wasting compute resources. This is algorithmically shown in Alg.2 (appendix C). It is studied empirically in table 1, setup in appendix B.3.
>  - The problem of §3.3 and §3.4 is illustrated in figure 1.
>  - For §3.3, our 'fix' is shown in figure 1 bottom-right and studied in the right of figure 4.
>  - §3.4 also improves path-degeneracy, the improvement is studied in table 1, setup in appendix B.3.
>
> Despite the details in the paper, we have included the table below for your convenience that summarizes the performance detriment of turning off each modification for our TRT-SMC shown in Figure 2. Note, that due to limited time and resources within this rebuttal period, we could only show this on RubiksCube for 10m training steps with the same statistical precision (30 seeds 99% intervals). Note that on this environment, and for this combination of hyperparameters, that some choices have less profound effect on the final performance than others. With the twisting having the strongest effect and revived resampling not differing statistically significantly. However, these effects can change as e.g., the depth of the SMC planner increases.
>
> | Rubikscube             | All      | w.o twisting | w.o policy backup | w.o. model-value | w.o revived resampling |
> | ---------------------- | -------- | ------------ | ----------------- | ---------------- | ---------------------- |
> | Solve rate $\hat{\mu}$ | $0.5835$ | $0.193$      | $0.5195$          | $0.5629$         | $0.5828$               |
>
> For the camera-ready, we are currently running experiments for this table on all environments.

---

### Official Review · Reviewer_ub84 · 2025-03-10

**Overall Recommendation:** 3

**Summary:**

The authors propose Trust-Region Twisted Sequential Monte Carlo (TRT-SMC) to improve the sample efficiency of SMC methods for reinforcement learning. One of the main motivations is that SMC methods scale better than Monte Carlo Tree Search (MCTS) because they enable higher GPU parallelization. The authors propose several changes to SMC: an adaptive trust region defined between the prior and greedy policies, revived terminal states with resampling, and some other improvements to policy and value estimation. TRT-SMC is compared against PPO, A2C, Gumbel MCTS, and SMC in 4 Brax/Jumanji environments, where it is shown to achieve high sample efficiency and (except for SMC) faster runtime.

## Update after rebuttal

I thank the authors for their rebuttal and follow-up discussion. In anticipation of the A2C results and inclusion of the $\alpha$-ablations that we discussed, I have raised my score from 2 to 3.

**Claims And Evidence:**

Claims are supported with evidence, but evaluation criteria are sometimes questionable (see below).

**Essential References Not Discussed:**

None to my knowledge.

**Experimental Designs Or Analyses:**

- Some hyperparameter choices appear arbitrary and are never justified: e.g., using $\alpha=0.1$ for the trust region.
- Ablations are conducted on only 1 environment each (Table 1), and the environment choice varies across ablation. This again gives the impression that the results may have been cherry picked. It would be better to conduct the ablations on all environments and include them in the appendix.

**Methods And Evaluation Criteria:**

- It seems that only a very small subset of Brax and Jumanji were used for evaluation (Figures 2 and 3). Only 4 environments are reported (2 from each domain), which raises concerns of potential cherry picking. I could not find any justification for the choice of these 4 specific environments in the paper.
- The A2C results are unreliable. Only 3 seeds are used for the learning curves in Figure 2, which are apparently taken from a previous paper. No confidence intervals are shown, so these results are not statistically significant and should not be reported.

**Other Comments Or Suggestions:**

It is confusing to use $q^*$ for the policy, because $q^*$ is typically reserved for the optimal action-value function. Is there possibly another symbol that can be used?

**Other Strengths And Weaknesses:**

**Strengths**

Strong empirical results in a few Brax and Jumanji environments. TRT-SMC appears to slightly worsen runtime over SMC because of the additional features, but it achieves better sample efficiency in the tested environments. It also appears to be both faster and more sample efficient than Gumbel MCTS and PPO.

**Weaknesses**

Although the paper goes into great detail regarding the proposed improvements to planning and particle filtering, the description of the outer loop (which interacts with the environment) is largely missing. What is especially unclear to me is whether the transition model is assumed to known or be must be learned. I think it is the latter based on the discussion of variational SMC in Section 2.2, but it would useful for reproducibility to include this explanation or pseudocode.

**Questions For Authors:**

1. Could you clarify what is meant here? “Although we compare sample efficiency to the baselines, this is only for reference since we do not account for the additional observed transitions by the planner” (end of p. 6). Are you not counting some of the sample interactions with the environment?
1. I was really surprised to see that PPO is the slowest of the 4 methods you compared, when it is the only model-free algorithm. Why is that the case?

**Relation To Broader Scientific Literature:**

Great background discussion and related work. The contributions are clearly placed in the literature.

**Theoretical Claims:**

The core algorithm and the proposed improvements are all reasonable ideas. The rationale behind each design choice is strongly justified. Mathematical results are correct to the best of my knowledge.

---

> ### Author Rebuttal · Authors · 2025-04-01
>
> We thank the reviewer for their time and effort. We divided our rebuttal in two parts, firstly on the experiment design and the second part on answering your comments.
>
> ---
>
> **Choice of environments:**
>
> For the choice of environments, we wanted a mix of discrete and continuous control environments. For this, we tried to closely stick to the choice by Macfarlane et al. (2024) as we believe they achieve this mix well. It also allows us to accurately compare our SMC baseline results. However we replaced Sokoban with Snake to drastically scale down the experiment costs. Based on Bonnet et al. (2024), we believed Snake to be sufficient as it is a sparse reward environment that benefits from planning, and can be learned with a smaller model and fewer environment steps.
>
> Concerning the ablations, the environments in the ablations were chosen to 1) spread out to not overfit results on one single environment, 2) to scale down experiment cost where possible, and 3) to verify that our modification actually solves an issue of interest.
>
> This third point is important as on some environments the effect of e.g., $\alpha$ is more profound than others (see extra results below). For instance, sparse reward environments can benefit more from value-guidance in search. Just like how exploration papers often test their method specifically on Montezuma's revenge, we also want to test our modifications in settings where they are most needed. Showing **isolated** effects of our modifications in ablations while still having good performance across the board keeps the discussion self-contained.
>
> **Choice of $\alpha$**
>
> The specific choice of $\alpha=0.1$ we used in the main results (figures 1 and 2) seemed to be robust across environments (not tuned per environment). This value also mirrors common trust-region parameters of $\epsilon = 0.1$ used in TRPO, MPO, and SPO.
>
> **Extra results:** We did not have complete results for $\alpha$ ablations at the time of submission (only a few seeds) and have since ran experiments for $\alpha$ to a greater extent. These results can be found here for: [on discrete envs](https://www.dropbox.com/scl/fi/4syw502kk6sstt6xlxwcd/alpha_sensitivity_disc.png?rlkey=np5ffukyviuat6ft8j43nmeix&st=rwzepxn4&dl=0), [on continuous envs](https://www.dropbox.com/scl/fi/y1ygjj4ez5mzjb0ckoogm/alpha_sensitivity_cont.png?rlkey=6sdycmjr44v6hz73xbeuo4pu6&st=9vip5ugs&dl=0), and [aggregated across envs](https://www.dropbox.com/scl/fi/nf7wrdchu3zl1751cs5mm/alpha_sensitivity_combined.png?rlkey=c98sprlxesszepyekzciz25pz&st=irxuoi50&dl=0). The figures report final min-max normalized average returns on all environments (30 seeds). The different lines show different planning budgets of $N=K \cdot m$ where $K$ (particles) and $m$ (depth) with $K=m$, and small $\alpha$ means sticking to the prior $\pi_\theta$ and large $\alpha$ means a greedy policy $\hat{\pi}^*$ over the predicted $Q$ values.
>
> Similar to e.g., the clipping-ratio in PPO, different environments have different optimal $\alpha$ ($0.1$ for discrete, and $0.01$ for continuous). Overall $\alpha=0.01$ seems to work even better than what was used in our submission $\alpha =0.1$ (which we picked from preliminary testing). However, both of these values are robust across environments.
>
> Click here for supplementary results [folder](https://www.dropbox.com/scl/fo/hy8p1nalfduq1tqf2moaw/AJOYML8UEQy5Lb9UmYIHOfg?rlkey=v887m5zaab0hcx260cg8m5yq4&st=oa6h6xtq&dl=0)
>
> **A2C:**
>
> A2C is indeed a weak baseline that could be removed. But, we included it to contrast the difference in implementations so that readers can compare against results by the official jumanji code.
>
> ---
>
> **Answers to Questions:**
>
> 1) This means that the SMC and MCTS agents use more environment samples (planning budget) per 'real' environment step. We do planning with the real environment model. This mirrors having a highly accurate simulator. See also our response to Reviewer SRnR on Figure 2.
>
> 2) PPO is slower in figure 3 due to the bottleneck of neural network training and relative cheapness of the environment model. This means we can use SMC planning to cheaply draw more environment samples which can then 1) extract a more informative learning target and 2) generate more diverse data (Hamrick et al., 2021).
>
> **Outer-loop pseudocode:**
>
> We prepared pseudocode for the outer loop to extend our discussion on approximate policy iteration in the background (see our response to reviewer SRnR on how we intend to revise this).
>
> **On $q^\star$**
>
> Sadly, the convention of optimal state-action values coincides with variational inference for which $q^*$ is standard to the optimal variational distribution. So we opted to write value functions with capitals $Q$ (with a hat $\hat{Q}$ for approximations) and use small $q$ for variational distributions. It is a sacrifice.
>
> ---
>
> We hope our response addresses your concerns, and we are happy to discuss on any further concerns or questions, and explain the extra results in more detail.

---

> > ### Comment · Reviewer_ub84 · 2025-04-04
> >
> > Thanks to the authors for their rebuttal. Overall, this paper has interesting ideas, is well written, and could potentially be accepted, but the empirical evaluation remains as my biggest concern.
> >
> > I appreciate the new hyperparameter sweeps, but my main concern remains that the given subset of environments was handpicked for the benefit of the proposed algorithm. I cannot see a convincing argument for why certain environments were excluded, given that it would require no new setup and the algorithms are quick to run.
> >
> > ---
> >
> > A few more minor comments below:
> >
> > > Choice of $\alpha$. This value also mirrors common trust-region parameters of used in TRPO, MPO, and SPO.
> >
> > To be precise, $\alpha$ cannot "mirror" TRPO or the other methods because those trust regions use a fixed value $\epsilon$ to bound the KL divergence, unlike your method which multiplies $\alpha$ by the KL, right? Now that you have conducted a sweep of $\alpha$, this matters less, but my point is that these values cannot be simply transferred when their functionalities are so different.
> >
> > > Concerning the ablations, the environments in the ablations were chosen to 1) spread out to not overfit results on one single environment, ...
> >
> > If overfitting was a concern in the ablations, then more than 1 environment should have been tested for each one. The current results are not sufficient to establish that the observed phenomena are environment agnostic.
> >
> > > A2C is indeed a weak baseline that could be removed. But, we included it to contrast the difference in implementations so that readers can compare against results by the official jumanji code.
> >
> > Unless the results can be provided with the same statistical confidence as the other learning curves, then such a comparison is not meaningful in my opinion. I do think it would be better to remove A2C in the absence of more seeds.

---

> > > ### Author Response · Authors · 2025-04-07
> > >
> > > Thanks to the reviewer for the reply. We will take your valuable feedback into account in the revision of our paper.
> > >
> > > Please see our response to your comments.
> > >
> > > ---
> > >
> > > **A2C baselines:** Your concern for directly using the Jumanji results is valid, we will rerun the Jumanji authors' A2C code for 30 seeds.
> > >
> > > ---
> > >
> > > **On the choice of $\alpha$:**
> > > Your summary is correct.
> > > We used a common value, 0.1, of a different but similar parameter as a starting point, meaning smaller values induce more conservative policy improvement and larger values induce a more aggressive operator.
> > > This seemed to work quite well across environments, so we didn't tune it.
> > > Apologies for being unclear in our rebuttal.
> > > We completely agree that copying this value from a differing methodology does not map one-to-one to our method, and is likely suboptimal.
> > > Indeed, in the [additional results posted in our rebuttal](https://www.dropbox.com/home/icml25?preview=alpha_sensitivity_combined.png) we see better choices for $\alpha$ averaged over environments (e.g., 0.01).
> > >
> > > ---
> > >
> > > **On the ablations:**
> > >
> > > We would like to clarify the purpose of our ablation studies.
> > > In Section 3, we identified several issues of integrating SMC with RL and present our solutions.
> > > Through the ablation studies, we then want to show whether a particular modification actually solves an issue for which we propose it.
> > > For this, we need environments where we expect such issues to occur.
> > >
> > > The ablations did not intend to show how well our modifications improve the average performance across environments.
> > > For that, we agree that adding many more environments makes for a more convincing conclusion.
> > > However, this answers a different question than "do our modifications solve the issue for which they were designed."
> > >
> > > For the question "if our modifications bring benefits across environments", we believe our main results (Figure 2) already answer it but taken together (not disentangled per modification). However, we also see the value in answering this question for each modification. Given our resources, we had to prioritize the other question as we believe it is more important, also our current experiments already took multiple GPU years to finish.
> > >
> > > ---
> > >
> > > **On choice of environments (again):**
> > >
> > > To clarify, we excluded Sokoban because of costs.
> > > We mentioned this in the rebuttal, but had to be brief due to space.
> > > To give more details, Sokoban requires at least ~5e8 environment steps to find a decent policy, which means ~5 GPU hours per run for our method. Note that this excludes the baselines, which can be more expensive (e.g. MCTS), and ablation studies. In contrast, Snake is much cheaper, requiring only ~15 GPU minutes per run.
> > >
> > > Finally, note that the Jumanji suite is extremely broad: it contains multi-agent tasks, tasks with exponentially scaled rewards, and tasks with highly specific observation and action factorizations and modalities.
> > > This requires **per-environment** tuning and different **neural network architectures**.
> > > In short, this means we do need additional setup.

---

### Official Review · Reviewer_SRnR · 2025-03-13

**Overall Recommendation:** 3

**Summary:**

Trust-Region Twisted Policy Improvement develops a new algorithm for performing Sequential Monte-Carlo (SMC) inspired by Monte-Carlo Tree Search (MCTS) for performing online planning in RL algorithms, which scales in performance with respect to the number of particles sampled during each planning step. The algorithm modifies and adapts several aspects of prior SMC approaches to be more RL specific - namely 1) fixing the path degeneracy problem through approximate message-passing  2) "twisting" the sampling by mixing $\exp Q$ with the current policy $\pi$ tradeoff sampling variance for bias  3) modifying value targets to use the values estimated by the search 4) revived resampling of absorbed states to avoid wasting compute on trapped particles

**Claims And Evidence:**

The following major claims are well supported in this paper:
* Planning with search is an important problem in RL (citations from Silver et al., 2018, Ye et al., 2021, Fawzi et al., 2022; Mankowitz et al., 2023.)
* Current bootstrapped particle-filtering methods suffer from several issues, including data-inefficiency and path degeneracy (Figure 1)
* TRT-SMC improves upon previous methods by addressing these issues (Figure 2, Figure 3, Section 5).

**Essential References Not Discussed:**

N/A

**Experimental Designs Or Analyses:**

Yes, the experiments appear to be valid. All methods are trained over 30 seeds, which is sufficient for establishing statistical significance, and hyperparameters and code package version are extensively documented in Appendix B. However, the curves can be misleading as the planning budget is not taken into account, exaggerating the sample efficiency of TRT-SMC and other SMC methods. It would be better to include this in the caption of Figure 2 so that readers who skip the authors' disclosure of this in the main text will not be mislead. Additionally, it would be good to include figures on the performance against "All Samples" (including online planning samples) rather than just "Training Samples". The TRT-SMC results will not look so good under this x-axis, but it is essential to include as total sample-efficiency is one of the primary concerns in RL.

**Methods And Evaluation Criteria:**

Yes. They performed experiments on the Brax continuous control tasks (Freeman et al., 2021) and Jumanji discrete environments (Bonnet et al., 2024), both of which are good RL test suites.

**Other Comments Or Suggestions:**

The control as inference part of the background can be shortened, as Levine 2018 already has an excellent tutorial on this. It would be great to see a longer exposition of SMC and MCTS, as this is the primary focus of the paper and the average RL reader will likely have much less experience with it.

**Other Strengths And Weaknesses:**

This paper addresses several important problems with existing SMC planners in RL, but does not propose any single novel component. This gives the paper a feeling of being a 'bag-of-tricks', where several somewhat small things were improved upon to stack together a better method. I think this style of research is good and useful, thus I advocate for accept. However, I will flag that the novelty is limited.

**Questions For Authors:**

N/A

**Relation To Broader Scientific Literature:**

This paper adapts the particle filter planner for deep reinforcement learning by addressing issues in variational sequential Monte Carlo that affect policy inference. The authors introduce TRT-SMC, which reduces path degeneracy and optimizes data efficiency, leading to improved runtime and *training* sample efficiency over traditional methods. They aim to inspire further integration of planning and Bayesian inference tools to enhance reinforcement learning algorithms.

**Theoretical Claims:**

I did not carefully check the proofs.

---

> ### Author Rebuttal · Authors · 2025-04-01
>
> We thank the reviewer for their time, effort and, accurate summary of our paper.
>
> Regarding limited novelty, we understand the concern but would also like to point to a slightly different perspective. With the benefit of hindsight it is easy to dismiss contributions by being reductive. For instance, the SMC-planner by Piche (2018) combined soft-actor critic with a policy extracted from a bootstrapped particle-filter, without significant changes to this filter. An overly reductive summary would say that they simply 'copy-paste' a sequential importance resampler for policy inference into the RL-loop. A similar thing can be stated about SPO by Macfarlane (2024) which replaced the MCTS-planner in AlphaZero with SMC, and then merged this into the MPO framework. There are always components being rehashed, but just like these prior works, it requires deep understanding to identify which parts of an algorithm work or don't work well, and how to deal with this.
>
> We directly point out **non-trivial issues** concerning SMC-planning for RL that were not yet addressed, or overlooked, in the literature. Their identification requires **deep technical understanding** on integrating RL with SMC. They are non-trivial since applying a particle-filter for policy inference is fundamentally different to how the method should be used for conventional state-estimation, this introduces challenges that only emerge when integrating them. These problems relate to path-degeneracy, inefficient trajectory sampling, and wasteful data generation. We also directly provide solutions for these issues that work well and have sound theoretical grounding. We also believe that further study of this topic holds significant value to the broader RL community.
>
> ---
>
> **On figure 2:**
>
> We understand your suggestion to adjust the $x$-axis of figure 2 to account for the extra environment samples due to the planning budget. In this case, PPO and A2C would indeed perform much better in comparison to the other agents! For completeness, we added this figure [here](https://www.dropbox.com/scl/fi/l1oswifleo63zsm1x86mb/new_results_adj.png?rlkey=yq2vxtrftt56r9cb00cccnd7h&st=dxjl93m9&dl=0). However, this sidesteps the research question as we wrote at the start of section 5.
>
> We really want to study which method implements a stronger local policy improvement operator, for which we compare against baseline SMC and gumbel MCTS. These all use the exact same planning budget and number of environment samples. This setting assumes we have a highly accurate simulator, which is also standard in prior work (AlphaZero, SMC for RL, SPO).
>
> We will include the above 2 sentences to the figure 2 caption and emphasize it more in the main text of section 5 as to not mislead the reader. We will include the attached figure to the appendix.
>
> Click here for supplementary results [folder](https://www.dropbox.com/scl/fo/hy8p1nalfduq1tqf2moaw/AJOYML8UEQy5Lb9UmYIHOfg?rlkey=v887m5zaab0hcx260cg8m5yq4&st=oa6h6xtq&dl=0)
>
> ---
>
> **On the background:**
>
> In line with reviewers bnoX and ub84 we will add pseudocode of the outer-loop training (see [link to EM-pseudocode](https://www.dropbox.com/scl/fi/ktxtkcuq00syv0k7i7vew/outer-loop-pseudocode.png?rlkey=dwxroqk5xrl629v7a4oesye9c&st=07wviyru&dl=0)) either in Appendix C or in the main text depending on whether we have space for additional ablation results. We can 1) add a paragraph heading in the last paragraph of section 2.1 with '**Outer-loop**' to emphasize/ structure the approximate policy iteration part, and 2) extend this text on how the estimated posterior policies $\hat{q}_t$ are used within this Expectation-Maximization loop in combination with neural network training. Section 2.2 and section 3 then continue in making improvements to the E-step approximation of $\hat{q}_t$.
>
> If needed, we can shorten the background, however we do need the definitions of the posterior-policy and value functions. Furthermore, the property of theorem 2.2 leads into how we motivate the exponentially twisted proposals. We can definitely shorten the formulas and move part of this derivation to the appendix (e.g., equations 2 and 3 can be shortened to 1 line) so that the proposed pseudocode and guiding text fit.

---

> > ### Comment · Reviewer_SRnR · 2025-04-08
> >
> > I thank the authors for their thoughtful response. Ensure to make the edits that are promised in the rebuttal, especially the new figure comparing total sample complexity.

---

### Official Review · Reviewer_bnoX · 2025-03-16

**Overall Recommendation:** 3

**Summary:**

- The paper creates a framework to apply sequential Monte Carlo planners to reinforcement learning settings by improving data generation through constrained action sampling and explicit terminal state handling,
- Specifically, SMC methods traditionally estimate the trajectory distribution under an unknown policy, rather than the specific unknown policy at the start of the planning, which causes high-variance estimation mismatches leading to a "trust-region twisted SMC" (TRT-SMC); so too reduce the estimates' variance, the paper uses exponential twisting functions to improve sampling trajectories inside the planner,
- The paper proposes a (to the best of my knowledge) novel method to deal with particles stuck in terminal states using a particular resampling strategy, and
- Finally, the paper demonstrates a better sample efficiency and runtime scaling through experiments in continuous control tasks and ablation studies.


## update after rebuttal

I have raised my score following the author's response.

**Claims And Evidence:**

- The paper rehashes a theoretical insight from Levine (2018) to prove theorem 2.2,
- The paper does not make any other formal theoretical claims, but rigorously justifies the algorithm design for the MCTS process in Algorithm 1

**Essential References Not Discussed:**

- I am not well versed with the new line of literature in this field.

**Experimental Designs Or Analyses:**

I did not check the experimental design in detail.

**Methods And Evaluation Criteria:**

The methods seem reasonable.

**Other Comments Or Suggestions:**

None

**Other Strengths And Weaknesses:**

- The paper is exceptionally well written,
- It is unclear why the result is novel, however, as there does not seem to be a technical difficulty underpinning the result

**Questions For Authors:**

I'd appreciate an answer to the point above.

**Relation To Broader Scientific Literature:**

The paper provides a nice approach to make parallel TRPO possible.

**Theoretical Claims:**

I checked the proofs of correctness of the theoretical claims, but it seems to mostly re-hash existing results from Levine (2018).

---

> ### Author Rebuttal · Authors · 2025-04-01
>
> We thank the reviewer for their effort and accurate summary of our paper. We address your points below.
>
> ---
>
> **Theoretical results:**
>
> Our paper does not introduce new theoretical results, **nor do we claim so**. It is correct that theorem 2.2 is known, it also goes quite farther back than Levine (2018), however this paper has a nice presentation on the theory. We restated this property since, as you also write, to motivate how we use the sequential Monte-Carlo planner and our proposed changes.
>
> The theorem shows that 1) posterior-policy estimation implies a regularized policy improvement, and 2) that learning on samples of this posterior-policy should maximize this lower-bound. We used the former insight to impose trust-regions on the proposal distribution in the SMC-planner and the latter to perform neural network training in the outer loop.
>
> **We believe that motivating our new method with known prior results should not merit rejection but instead strengthen our approach.** But, could you elaborate whether the critique is that we put too much focus on the background? If so, we will shorten this part.
>
> ---
>
> **Response to Technical difficulty:**
>
>  > It is unclear why the result is novel, however, as there does not seem to be a technical difficulty underpinning the result
>
> It is unclear whether you refer to the theory, our empirical results, or our method.
>
> As mentioned above, the theory is there to support our modifications and show how SMC planning works (e.g., the importance-sampling weights). We don't understand why supporting a newly proposed method with known previous results should be a basis for rejection.
>
> Regarding our method, we discovered and described non-trivial issues in current approaches for SMC based planning in RL, and present concrete approaches on how to deal with this. Prior work overlooked these issues, and their identification requires **deep technical understanding** on integrating RL with SMC.
>
> Also note that simplicity (if that is the concern) of a new method does not detract from its novelty, instead it should improve its clarity and broader applicability.
>
> Perhaps you could clarify your position?
>
> ---
>
> **Minor comments:**
>
> Our method is technically not a parallel TRPO (cf. Fickinger et al. 2021). The main distinctions are:
>  1) The KL-divergence constraint: the arguments to this constraint would be swapped for TRPO vs. our approach (MPO-based; theorem 2.2).
>  2) TRPO can already be parallelized, simply by increasing the number of asynchronous environments.
>  3) There is a fundamental difference in how data is collected and how policy learning is performed -- relating back to our previous comment, we can extend the outer-loop discussion in the background to make this distinction more clear.
>
> ---
>
> **References**
>
> Fickinger, A., Hu, H., Amos, B., Russell, S., \& Brown, N. (2021). Scalable online planning via reinforcement learning fine-tuning. *Advances in Neural Information Processing Systems*, *34*, 16951-16963.

---

### Decision · Program_Chairs · 2025-05-01

**Decision:**

Accept (poster)

**Comment:**

The proposed TRT-SMC algorithm aims to enhance Sequential Monte Carlo (SMC) methods for RL by integrating several techniques targeting issues like path degeneracy and sample efficiency. Reviewers found the constituent ideas reasonable and acknowledged the strong empirical results shown in limited test environments. However, significant concerns were raised regarding the work's novelty, often described as a "bag-of-tricks," and the rigor of the experimental evaluation, particularly the small number of environments and insufficient ablation studies, suggesting potential cherry-picking. Despite these major reservations, the consensus leaned towards weak acceptance, recognizing the potential practical value.